# IFI16, a nuclear innate immune DNA sensor, mediates epigenetic silencing of herpesvirus genomes by its association with H3K9 methyltransferases SUV39H1 and GLP

Arunava Roy[1]*, Anandita Ghosh[1], Binod Kumar[2], Bala Chandran[1]*

[1]Department of Molecular Medicine, Morsani College of Medicine, University of South Florida, Tampa, United States; [2]Department of Microbiology and Immunology, Chicago Medical School, Rosalind Franklin University of Medicine and Science, North Chicago, United States

**Abstract** IFI16, an innate immune DNA sensor, recognizes the nuclear episomal herpes viral genomes and induces the inflammasome and interferon-β responses. IFI16 also regulates cellular transcription and act as a DNA virus restriction factor. IFI16 knockdown disrupted the latency of Kaposi's sarcoma associated herpesvirus (KSHV) and induced lytic transcripts. However, the mechanism of IFI16's transcription regulation is unknown. Here, we show that IFI16 is in complex with the H3K9 methyltransferase SUV39H1 and GLP and recruits them to the KSHV genome during de novo infection and latency. The resulting depositions of H3K9me2/me3 serve as a docking site for the heterochromatin-inducing HP1α protein leading into the IFI16-dependent epigenetic modifications and silencing of KSHV lytic genes. These studies suggest that IFI16's interaction with H3K9MTases is one of the potential mechanisms by which IFI16 regulates transcription and establish an important paradigm of an innate immune sensor's involvement in epigenetic silencing of foreign DNA.

DOI: https://doi.org/10.7554/eLife.49500.001

*For correspondence:
arunavaroy@usf.edu (AR);
chandran@usf.edu (BC)

**Competing interests:** The authors declare that no competing interests exist.

## Introduction

Entry of foreign DNA into the nucleus of a eukaryotic cell triggers numerous innate immune responses such as the induction of the inflammasome, type 1 interferon and DNA damage responses, as well as the recruitment of the ND10 or PML body components. In addition to the secretion of pro-inflammatory cytokines as a defense, the cell also maintains its homeostasis by limiting the expression of the invading foreign genes. Transcription factors like Daxx, Sp1 and PML (*Poleshko et al., 2008*; *Merkl et al., 2018*), and heterochromatin inducing factors like HDAC1 and CBX3 (*Poleshko et al., 2008*) have been shown to be instrumental in silencing foreign viral DNA. Interestingly, a Pyrin-Hin200 family nuclear innate immune DNA sensor, IFI16 (Interferon-γ-inducible protein 16) has emerged as a preeminent viral transcription restriction factor against a number of DNA viruses (*Gariano et al., 2012*; *Orzalli et al., 2013*; *Merkl and Knipe, 2019*; *Johnson et al., 2014*; *Lo Cigno et al., 2015*). Although the innate immune DNA sensor role of IFI16 where it senses foreign (*Kerur et al., 2011*; *Orzalli et al., 2013*) or damaged self DNA (*Ouchi and Ouchi, 2008*) and induces the ASC-dependent inflammasome pathway (*Kerur et al., 2011*; *Ansari et al., 2013*; *Dutta et al., 2015*) and the IFN-β pathway through the STING-TBK1-IRF3 axis (*Unterholzner et al., 2010*; *Iqbal et al., 2016*) is well understood, its transcription restriction role is not fully understood.

*Gariano et al. (2012)* observed that IFI16 restricted human cytomegalovirus (HCMV) transcription and replication by displacing the Sp1 transcription factor. *Orzalli et al. (2013)*, and *Merkl and Knipe (2019)* showed that IFI16 promotes silencing of human herpes simplex virus type-1 (HSV-1) gene expression and replication by adding the repressive chromatin mark, H3K9me3. Our studies showed that IFI16 restricts HSV-1 replication by modulating the binding of H3K4me3 and H3K9me3 histone marks and the RNA Pol II, TATA-binding protein (TBP) and Oct1 transcription factors (*Johnson et al., 2014*). Similar observations were made by *Lo Cigno et al. (2015)* for human papillomavirus 18 (HPV18).

Recently, we observed that IFI16 is essential for the repression of Kaposi's sarcoma associated herpesvirus (KSHV) lytic gene transcription during latency and thus facilitates latency maintenance (*Roy et al., 2016*). IFI16 knockdown (KD) in the latently KSHV-infected primary effusion B-lymphoma (PEL) BCBL-1 and BC-3 cell lines resulted in a global increase of KSHV lytic transcripts, proteins, and viral genome replication but not latent genes. We also demonstrated these results during KSHV lytic cycle induction in TREX-BCBL-1 cells with the doxycycline-inducible lytic cycle switch replication and transcription activator (RTA) gene. Overexpression of IFI16 reduced lytic gene induction by a chemical agent TPA. Intracellular viral genome copy number and virion particle associated KSHV DNA copy numbers were also elevated as a result of this KD. IFI16 chromatin immunoprecipitation assays (ChIP) showed that IFI16 binds to the promoters of all temporal KSHV gene classes. In addition, IFI16 repressed the transcription of KSHV luciferase promoter constructs in the uninfected epithelial SLK and osteosarcoma U2OS cells. Furthermore, during lytic reactivation of KSHV, we observed that IFI16 was polyubiquitinated and degraded via the proteasomal pathway, and thus relieving the lytic promoters of the repressive action of IFI16 to create a conducive environment for lytic gene expression. Blocking of KSHV DNA replication and late lytic gene expression resulted in the absence of IFI16 degradation. Our studies suggested that KSHV utilizes the innate immune nuclear DNA sensor IFI16 to maintain its latency by repressing the lytic gene transcription.

The transcription regulatory role of IFI16 has been also reported in other unrelated systems (*Johnstone et al., 1998*; *Caposio et al., 2007*; *Kang et al., 2014*). *Johnstone et al. (1998)* fused IFI16 to GAL4 DNA binding domain (DBD) and observed that it acts as a potent transcription repressor when positioned in proximity to a GAL4DBD binding sequence containing promoter. *Caposio et al. (2007)* reported that overexpressing IFI16 resulted in an increased expression of genes involved in immunomodulation, cell growth, and apoptosis. IFI16 has been shown to interact with cellular transcription factors such as SP1 (*Luu and Flores, 1997*; *Lo Cigno et al., 2015*) and p53 (*Johnstone et al., 2000*; *Kwak et al., 2003*). *Kang et al. (2014)* found that IFI16 is associated with the promoter of the estrogen receptor α (ERα) gene ESR1 and plays a role in the transcriptional regulation of ESR1 gene in breast cancer (*Kang et al., 2014*). *Thompson et al. (2014)* described a regulatory role for IFI16 in the transcriptional regulation of IFN-α gene expression. Recently, a closely related murine PYHIN family member p205 has been shown to control ASC gene expression by regulating RNA polymerase II binding to its promoter (*Ghosh et al., 2017*).

In this study, we used KSHV latency establishment and maintenance as a model system to study the transcription and possible epigenetic modulatory roles of IFI16. KSHV, an oncogenic human herpesvirus, is etiologically associated with endothelial Kaposi's sarcoma (KS), primary effusion B-cell lymphoma (PEL) and B-lymphoproliferative multicentric Castleman's disease (MCD) (*Chang et al., 1994*; *Cesarman et al., 1995*; *Soulier et al., 1995*). Like all other herpesviruses, KSHV establishes a lifelong latency in humans with periodic lytic reactivation and reinfection (*Boshoff and Chang, 2001*). Based on the temporal regulation of expression, the lytic genes are classified as immediate early (IE), early (E) and late (L) genes. After primary infection of permissive cells, the chromatin-free input KSHV genome enters the nucleus and undergoes rapid chromatinization which is mediated by cellular epigenetic factors and viral chromatin regulatory elements (*Günther and Grundhoff, 2010*; *Toth et al., 2013b*; *Günther et al., 2014*). This chromatinized viral genome is maintained as an extrachromosomal episome. During latency, lytic genes are under tight transcriptional repression by epigenetic factors and transcriptional repressors (*Krishnan et al., 2004*). Upon conducive conditions such as hypoxia, infection by other pathogens, inflammatory cytokines, or immune suppression, KSHV transitions from latency to lytic reactivation leading to progeny virion formation (*Toth et al., 2013b*). KSHV lytic reactivation can be induced by treating latently infected cells in culture with small molecule inhibitors of epigenetic modulators such as histone methyltransferases (HMTases), histone

deacetylases (HDACs), histone acetyltransferases (HATs) and DNA methyltransferases (DNMTs) (*Hopcraft et al., 2018*).

The mechanisms of KSHV genome maintenance during latency are under intense investigations. *Toth et al. (2010)* reported that activating histone marks AcH3 and H3K4me3 colocalized on the KSHV genome and predominantly occupied the latency and the early-lytic regions during latency. On the other hand, the repressive H3K27me3 mark was widely distributed throughout the KSHV genome, while H3K9me3 was restricted mainly to two regions spanning 30–60 kb and 95–115 kb encoding several late lytic genes (ORF16–40 and ORF58–68). In agreement with Toth et al.'s results, *Günther and Grundhoff (2010)* also found H3K9me3 to be mainly restricted to two consecutive KSHV genomic regions, ~33–46 kb and 100–114 kb in latently infected cells. Both these authors found that many lytic loci which are transcriptionally inactive during latency are occupied by both activating as well as repressive histone marks, instigating the hypothesis that these promoters are 'bivalent' and are suspended in a transcriptionally poised state capable of reactivating rapidly when needed (*Toth et al., 2013a*; *Ter Horst and Luiten, 1987*; *Günther and Grundhoff, 2010*). More recently, *Sun et al. (2017)* performed ChIP-Seq experiments on classic KS tissues and found that the activating acetylated H3 (H3Ac) marks were restricted to the latency locus, while the repressive H3K27me3 was widespread on KSHV genome.

Although the transcription repression and the possible epigenetic histone mark modification roles of IFI16 are known, no definitive mechanism has been identified till date linking IFI16 to epigenetic chromatin remodeling. Here, we used KSHV latency as a model and asked the question - 'how does IFI16 modulate epigenetic histone marks leading to its transcription regulatory activity?'. Our studies show that KD of IFI16 most significantly reduces H3K9me3 and increases H3K4me3 on the KSHV gene promoters, and that IFI16 interacts with H3K9 histone methyltransferase (H3K9-MTase) Suppressor of variegation 3–9 homolog 1 (SUV39H1) and G9a-like protein (GLP) both in uninfected and infected cells. IFI16 KD and knockout (KO) drastically reduces the recruitment of these two H3K9-MTase onto the KSHV genome which was confirmed by the IFI16 rescue experiments in KO cells. Our studies suggest that IFI16 dependent recruitment of GLP mono and di-methylates (me1/me2) at H3K9 on the KSHV genome, while SUV39H1 further establishes tri-methylated H3K9me3 marks. This consequently regulates the recruitment of heterochromatin protein 1-α (HP1α) protein which functions downstream of H3K9me3 resulting in DNA compaction and heterochromatization of the KSHV genome and silencing of the KSHV genes, especially the late lytic genes. Our findings unravel a previously unknown function of IFI16 namely the interactions with SUV39H1 and GLP H3K9-MTases that facilitate the epigenetic silencing of foreign viral DNA.

## Results

### Knockdown of IFI16 causes prominent changes in the deposition of H3 lysine methylation marks and RNA Pol II on KSHV promoters

IFI16 suppresses the transcription from KSHV lytic promoters and KD of IFI16 results in lytic reactivation of latently infected cells (*Roy et al., 2016*). To investigate the potential mechanism(s) of IFI16-mediated suppression of KSHV lytic transcription, we determined the role of IFI16 in recruiting the five known H3 lysine trimethylation marks (H3Kme3) - H3K4me3, H3K9me3, H3K27me3, H3K36me3 and H3K79me3, on the KSHV genome. Among these, H3K4me3, H3K36me3, and H3K79me3 are known to be associated with transcriptionally permissive euchromatin, whereas H3K9me3 and H3K27me3 are known to be associated with transcriptionally repressive heterochromatin. For this, we depleted IFI16 in latently infected BCBL-1 cells using lentivirus-mediated shRNA (shIFI16) transduction and evaluated the recruitment of the different H3Kme3s and total H3 by chromatin immunoprecipitation (ChIP) after 72 hr of transduction. Compared to control shRNA (shC), we observed ~70% KD of IFI16 as evaluated by qRT-PCR and WB (*Figure 1A and B*, respectively). Since IFI16 KD results in the induction of KSHV lytic genes (*Roy et al., 2016*), when we evaluated the levels of lytic ORF50 mRNA by qRT-PCR, we observed a six-fold increase after IFI16 KD (*Figure 1A*).

In addition to the five H3Kme3s, we also performed ChIP against RNA pol II to determine whether IFI16 KD results in reduced RNA Pol II mediated transcription. For real-time PCR analysis of the ChIP DNA, we used four KSHV promoter primers representing the four temporal KSHV gene classes, namely, pORF73 (La), pK8 (IE), pvIRF2 (E), and pORF63 (L). To better represent the resulting

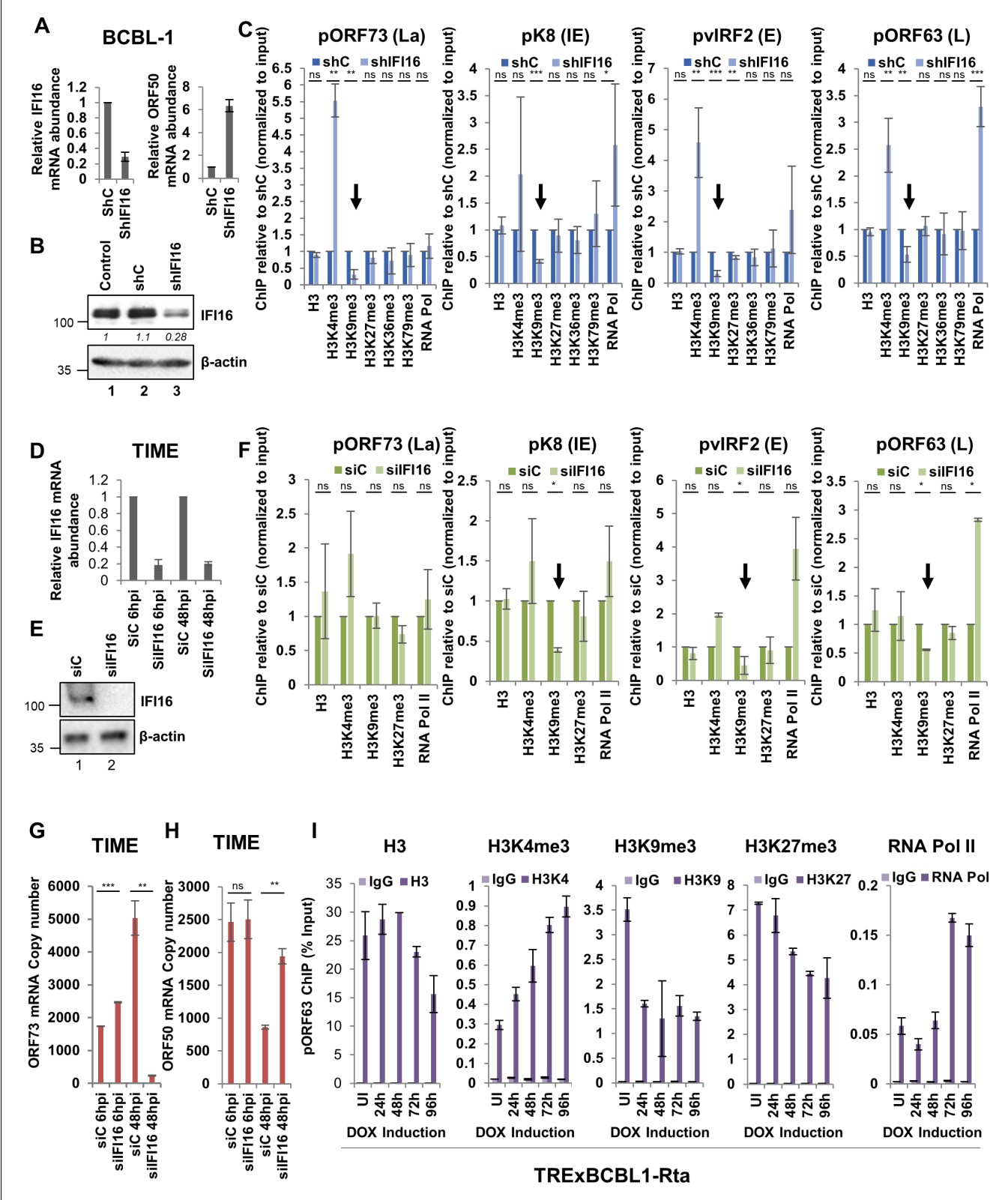

**Figure 1.** Effect of IFI16 knockdown (KD) on H3K9me3 and RNA Pol II deposition on KSHV lytic gene promoters. (**A**) IFI16 was KD in BCBL-1 cells using shRNA lentivirus for 72 hr and KD efficiency was assessed by q-RT PCR and successful induction of lytic KSHV ORF50 gene as a result of IFI16 KD was assessed by q-RT PCR of ORF50. (**B**) WB showing IFI16 KD compared to untreated or shC treated BCBL-1 cells. (**C**) ChIP was performed after lentivirus-mediated IFI16 KD in BCBL-1 cells or shC-BCBL-1 cells. Deposition of different histone H3 lysine tri-methylation marks (H3, H3K4me3, H3K9me3,

*Figure 1 continued on next page*

*Figure 1 continued*

H3K27me3, H3K36me3 and H3K79me3) and RNA Pol II on four different KSHV promoters (pORF73- La, pK8- IE, pvIRF2- E, and pORF63- L) representing the four different temporal KSHV gene classes were tested by q-PCR. ChIP efficiencies normalized to input chromatin are shown as relative to shC control. (D - H) TIME cells were electroporated with either siC or siIFI16. After 72 hr, cells were de novo infected with KSHV (100 DNA copies/cell) for 6 or 48 hr. IFI16 KD efficiencies were assessed by q-RT PCR of the IFI16 gene (D) and WB of IFI16 (E). (F) ChIP was performed after 48 hr of de novo infection of IFI16 KD TIME. (G and H) q-RT PCR (one step TaqMan) of KSHV latent ORF73 (G) and lytic ORF50 (H) mRNA expression normalized to cellular RNaseP after 6 and 48 hr of de novo infection of TIME cells previously treated with siIFI16 of siC for 72 hr. (I) Lytic cycle was induced in TRExBCBL1-RTA cells using doxycycline. At 0, 1, 2, 3 and 4 days post-induction, ChIP was performed. Deposition of different H3 lysine tri-methylation marks (H3, H3K4me3, H3K9me3 and H3K27me3) and RNA Pol II on the ORF63 promoters was tested by q-PCR. ChIP with control IgG was also performed. ChIP efficiencies are represented as % input. Data shown are averages of the results of at least three experiments ± SD. *=p < 0.05; **=p < 0.01; ***=p < 0.001 (unpaired t test).

DOI: https://doi.org/10.7554/eLife.49500.002

fold changes, ChIP efficiencies were normalized to input chromatin and represented as relative to shC. Among all the H3Kme3s tested, we observed that IFI16 KD results in a consistent increase in H3K4me3 recruitment and decrease in H3K9me3 recruitment (black arrows) on all the KSHV promoters tested (*Figure 1C*). Total H3 remained relatively unaltered after IFI16 KD, and in contrast, RNA Pol II recruitment was enriched on pK8, pvIRF2 and pORF63 but not pORF73 (*Figure 1C*). This observation is in agreement with our previous observations where we demonstrated that IFI16 KD in BCBL1 cells had little effect on latent gene transcription, while lytic genes of all gene classes were induced (*Roy et al., 2016*).

Next, we tested the effect of IFI16 KD during de novo KSHV infection of TIME cells, which are human neonatal foreskin microvascular endothelial cells immortalized with telomerase reverse transcriptase (hTERT). These cells exhibit a normal karyotype, extended lifespan in culture, and endothelial characteristics at late passages which make them an ideal model for KSHV de novo infection. These cells were electroporated with a pool of siIFI16 or siC and 72 hr later, infected with KSHV at 100 DNA copies/cell for 6 or 48 hr. qRT-PCRs (*Figure 1D*) and WBs (*Figure 1E*) assessed IFI16 KD efficiencies. Using the same ChIP promoter primers, we found that 48 hr post-infection (p.i.) with KSHV in IFI16 depleted cells, recruitment of H3K9me3 on lytic pK8, pvIRF2 and pORF63 decreased at least by 0.5-fold (black arrows) while H3K4me3 and H3K27me3 did not change significantly. Similar to our observations in BCBL-1 cells, recruitment of RNA Pol II increased significantly on these promoters, but not on latent pORF73 (*Figure 1F*).

To further assess if IFI16 KD can influence the outcome of KSHV de novo infection, we measured the copy numbers of latent ORF73 and lytic ORF50 mRNAs after IFI16 KD in TIME cells (*Figure 1G and H*). In the control siC-treated cells, ORF73 mRNA increased between 6 hr and 48 h p.i. and ORF50 mRNA decreased by about 2.5-fold after its initial round of transcription during the early phases of de novo infection. This is in agreement with findings by *Krishnan et al. (2004)* where they showed the transient transcription of a subset of KSHV lytic genes in addition to the latent genes, with roles in latency establishment immediately after de novo infection of HMVEC-d cell. In the IFI16 KD cells, although no major changes were observed at 6 h p.i., ORF73 mRNA decreased significantly at 48 h p.i. compared to siC while ORF50 mRNA increased by about 2-fold. This suggested that when IFI16 is depleted, KSHV fails to establish latency and lytic phase commences. This corroborates with our previous observations in PEL cells where we demonstrated that IFI16 is important for the maintenance of KSHV latency and in its absence, KSHV reactivates to lytic life cycle (*Roy et al., 2016*).

To determine the importance of H3 lysine methylations in KSHV latent and lytic cycles, we determined the dynamics of different H3Kme3s on the KSHV genome during reactivation from latency to lytic cycle. TRExBCBL1-Rta cells carry latent KSHV genomes and an epitope-tagged KSHV lytic cycle switch replication and transcription activator (RTA) protein cassette under the control of a tetracycline-inducible promoter. We chose these PEL cells as KSHV lytic cycle can be reactivated by doxycycline (DOX) induced RTA expression (*Nakamura et al., 2003*) instead of phorbol esters and sodium butyrate, both of which are known to effect transcription globally and are not KSHV specific. We induced the TRExBCBL1-Rta cells with doxycycline and evaluated the recruitment of total H3 and different H3Kme3s on the late lytic ORF63 promoter by ChIP at 1, 2, 3 and 4 days post-induction. The results were represented as % of input to confirm the ChIP efficiencies of the respective antibodies,

and we also included IgG ChIP (*Figure 1I*). We have previously reported that IFI16 is polyubiquitinated and degraded via the proteasomal pathway as soon 48 hr post-induction of TREXBCBL-1 cells (*Roy et al., 2016*). We observed that recruitment of total H3 did not change between 1- and 3 days post-induction but decreased at 4 days post-induction. This possibly represents the newly synthesized viral genomes that are not chromatinized (*Oh and Fraser, 2008*). Recruitment of H3K4me3 on the late ORF63 promoter increased steadily until day 4, suggesting the gradually increasing transcriptional activity from this promoter. H3K9me3 recruitment decreased sharply on day 1 and remained at similar levels until day 4. This emphasizes the repressive role of H3K9me3 during lytic reactivation. H3K27me3 recruitment also decreased but more gradually between day 1 and day 4. RNA Pol II was predominantly recruited between days 3 and 4 which was expected as ORF63 is a late gene.

Together, these observations suggested the following: a) IFI16 is important for the recruitment and maintenance of the repressive H3K9me3 mark and for the exclusion of the permissive H3K4me3 mark on the KSHV genome both, after de novo infection and during prolonged latency; b) IFI16 influences the recruitment of RNA Pol II on KSHV lytic promoters; c) depletion of IFI16 during de novo infection of endothelial cells results in lytic gene expression and failure to establish latency, and d) re-distribution of H3K9me3 along with H3K27me3 and H3K4me3 plays an important role during reactivation from latency.

## H3K9 methylations play an important role in KSHV life cycle and IFI16 interacts with cellular H3K9 methyltransferases (H3K9MTase)

Based on the previous studies of IFI16-mediated transcriptional repression and possible role of IFI16 in the modulation of H3K9me3 on herpesvirus genomes, we next determined the connection of this histone mark with IFI16. To test the importance of H3K9 methylations on KSHV life cycle, we used A-366, a potent and selective inhibitor of H3K9 me2/me3 methylation (*Kaniskan et al., 2018*). MTT toxicity assays revealed that 100 µM concentration of A366 is non-toxic on BCBL-1 cells for up to 72 hr (*Figure 2A*). BCBL-1 cells were treated for 72 hr with 10 µM or 100 µM concentrations of A366 or mock treated with DMSO vehicle control and we measured the expression of all four KSHV temporal gene by real-time qRT-PCR (*Figure 2B*). We observed that although there were no significant changes at 10 µM, treatment with 100 µM A366 induced robust initiation of lytic IE, E, and L gene transcripts. As a control, we also evaluated cellular IFI16 and GAPDH transcripts and found no significant changes after A366 treatment. When we tested the levels of different H3 lysine methylations after 72 hr 100 µM A366 treatment, we observed that A366 specifically inhibited H3K9me2 and H3K9me3 methylations, and the levels of IFI16 did not change significantly (*Figure 2C*). Recently, *Hopcraft et al. (2018)* screened a number of H3 MTases in an attempt to identify host chromatin-modifying proteins that are essential for maintaining KSHV latency and found that 1 µM and 10 µM concentrations of A366 were ineffective in inducing KSHV lytic cycle. Although this observation is comparable to our findings, these authors did not test 100 µM concentration of A366.

We next hypothesized that IFI16 binds, recruits and maintains specific H3K9 MTase(s) on the KSHV genome leading to the observed changes in H3K9me3 occupancy on the KSHV genome. To test this, we infected TIME cells with KSHV (100 DNA copies/cell) for 6 or 24 hr, isolated the nuclear fractions, treated with benzonase to digest all nucleic acids, and immunoprecipitated (IP) with anti-IFI16 antibody or isotype control IgG. The IPs were eluted under non-denaturing conditions to preserve the enzymatic activity of associated H3K9MTase and we performed a H3K9 methyltransferase activity assay (Materials and methods) with the eluate. We observed that IFI16 pulled down significant H3K9 MTase activity compared to the IgG control (*Figure 2D*) ranging between 0.4 and 0.5 ng/h/mg which were 10-fold lesser compared to the input lysates. KSHV infection did not alter the levels of intracellular MTases or their interaction with IFI16 as both uninfected and infected conditions showed comparable enzymatic activities.

Till date, nine H3K9MTase have been identified in eukaryotes (*Li et al., 2016*) and we were able to obtain reliable commercial antibodies against seven of them (SUV39H1, SUV39H2, GLP, G9A, SETDB1, SETDB2 and RIZ1). To identify the H3K9MTases interacting with the KSHV genome and are thus relevant in the context of KSHV life cycle, we infected TIME cells with 5-ethynyl-2´-deoxyuridine (EdU) genome labeled or unlabeled control KSHV (100 DNA copies/cell) for 24 hr followed by EdU-genome pulldown assay using Click chemistry (Material and methods). The pulldown eluates and their corresponding inputs were blotted for the presence of different H3K9MTases (*Figure 2E*).

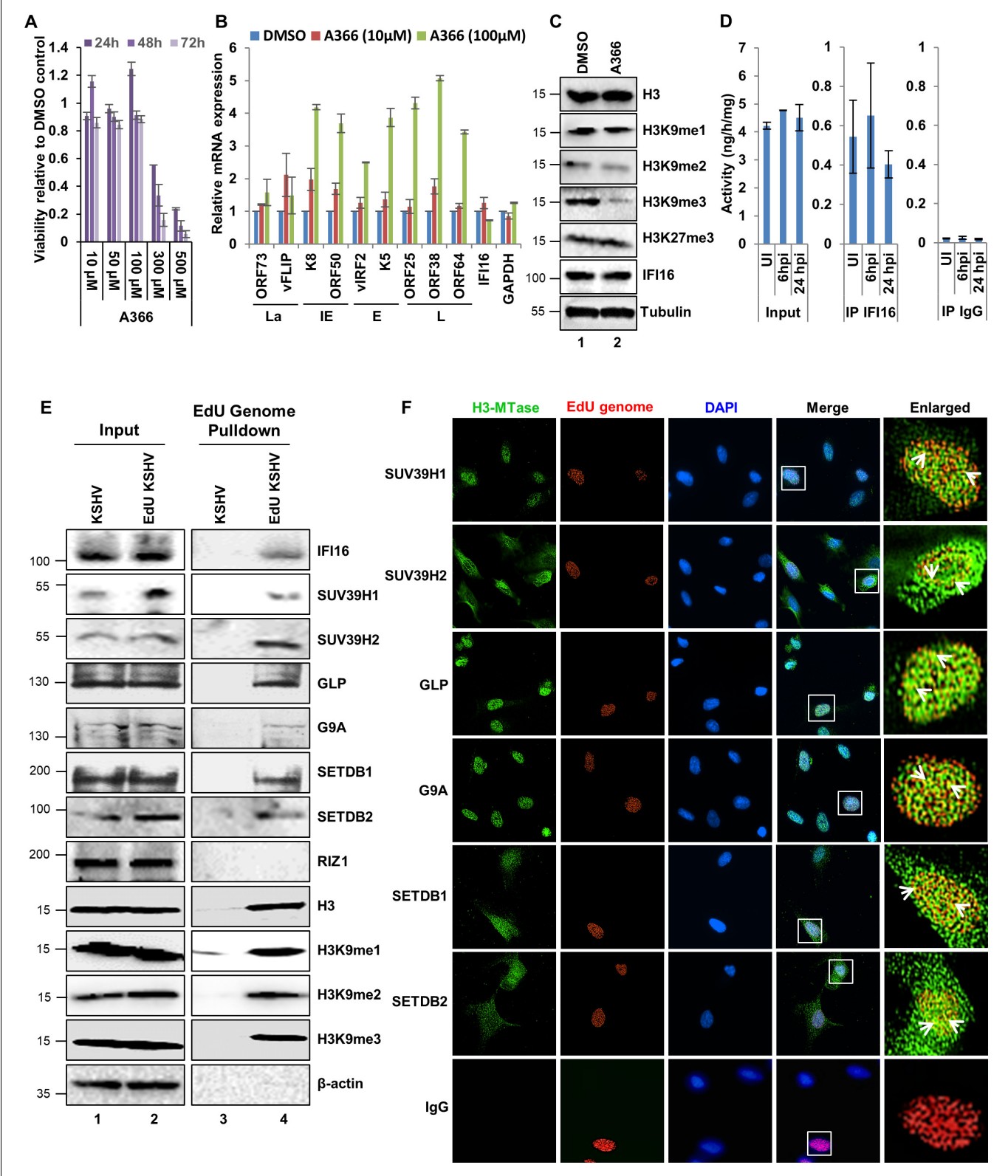

**Figure 2.** Effect of A366 on KSHV life cycle and the demonstration of IFI16's association with cellular H3K9 methyltransferase(s) (H3K9 MTase) and recruitment of various H3K9 MTases to the KSHV genome during de novo infection. (**A**) MTT cell viability assay of BCBL-1 cells treated with the H3K9me3 specific chemical inhibitor A366 at different concentrations and different time points. (**B**) q-RT PCR (two-step, sybr Green) of KSHV mRNAs in BCBL-1 cells treated for 72 hr with either vehicle control DMSO or A366 (10 μM and 100 μM). (**C**) WB of different H3 methylations and IFI16 after A366

*Figure 2 continued on next page*

*Figure 2 continued*

treatment of BCBL-1 cells. (D) H3K9 methyltransferase activity (ng/h/mg) assay. TIME cells were infected with KSHV for 6 or 24 hr followed by isolation of nuclear fraction, benzonase treatment and IP with anti-IFI16 or control IgG in the presence of benzonase using the catch and release method. Elution was performed under non-denaturing conditions to keep the associated H3K9 methyltransferase active. H3K9 methyltransferase activity was assayed in the eluate (Materials and methods). *, p<0.05; **, p<0.01; ***,<0.001; unpaired t-test. (E) TIME cells were infected with KSHV genome labeled with EdU or unlabeled control KSHV (100 DNA copies/cell) for 24 hr followed by EdU-KSHV genome pulldown using Click chemistry. The inputs and eluates were blotted for different H3K9 MTases. (F) TIME cells were infected with EdU-labeled KSHV as in (D) and stained using the Click-iT EdU Alexa Fluor 594 Imaging Kit (red). Subsequently, IFA was performed against different H3K9 MTases and colocalization of the IFA signal (green) with KSHV EdU-genome staining (red) resulting in yellow was evaluated (enlarged image, white arrows).
DOI: https://doi.org/10.7554/eLife.49500.003

Except for RIZ1, we could detect six of the seven H3K9 MTases tested – SUV39H1, SUV39H2, GLP, G9A, SETDB1, and SETDB2. In the same experiment, we also tested the recruitment of H3K9me1, me2, and me3 and observed that all three H3 marks are associated with the KSHV genome within 24 hr of de novo infection. IFI16 was used as a positive control. β-actin used as a negative control was not detected in the EdU genome pull-down.

To validate the association of these MTases with the KSHV genome, we performed immunofluorescence assay (IFA) for the six identified MTases on TIME cells infected with EdU-KSHV for 24 hr (*Figure 2F*). EdU-labeled genomes were detected using Click chemistry-based fluorescent staining (*Figure 2F*, red). All the six H3K9MTase (*Figure 2F*, green) colocalized with the EdU-KSHV genome (*Figure 2F*, yellow, white arrowheads) and thus confirming their association with the KSHV genome during de novo infection. SUV39H1, GLP, and G9A were detected predominantly in the nucleus, while SUV39H2, SETDB1 and SETDB2 were observed both in the nucleus and cytoplasm. IgG used as a negative control did not show any interactions. These results demonstrated that: a) treatment of BCBL-1 cells with A366 confirmed the importance of H3K9 methylations in KSHV gene regulations. b) IFI16 physically interacts with one or multiple H3K9MTases, and c) six different H3K9 MTases are associated with the KSHV genome.

## IFI16 interacts with H3K9 MTase SUV39H1 and GLP in KSHV latently infected cells

To identify the IFI16 interacting H3K9MTases, we performed co-immunoprecipitation (co-IP) experiments with benzonase-treated nuclear fractions from the latently infected BCBL-1 and BC-3 PEL cells and the uninfected BJAB lymphoma cells. IP of IFI16 resulted in efficient co-IP of SUV39H1 and GLP and to a lesser extent, G9A (*Figure 3A*, left panel, lanes 1–3). In contrast, all the other H3K9 MTases tested did not co-IP with IFI16. The H3K27 methylating PRC2 complex protein EZH2 was also tested but did not co-IP with IFI16. In addition, we also tested IFI16's ability to interact with H3, H3K9me3 and the heterochromatin binding protein HP1α, and none of them co-IPed with IFI16. ASC which is known to interact with IFI16 in PEL cells to form the inflammasome complex (*Singh et al., 2013*) was used as a positive control (*Figure 3A*).

LANA1, a KSHV latent protein, has been reported to recruit specific H3 methylating enzymes to the viral genome. *Sakakibara et al. (2004)* reported that LANA1 interacts with SUV39H1 leading to the accumulation of heterochromatin components on the TR sequence of the KSHV genome. They used DNA pull-down assay with a biotinylated DNA fragment that contained a LANA1-specific binding sequence and a maltose-binding protein pull-down assay to reach their conclusion. More recently, *Toth et al. (2016)* found that LANA1 binds and recruits EZH2 onto the KSHV lytic promoters leading to H3K27me3 mediated heterochromatin structure on the viral genome during de novo infection of SLK cells. To better understand LANA1's interactions with cellular H3K9 MTases under our experimental conditions, we conducted co-IP experiments with anti-LANA1 antibody. We observed that LANA1 interacts with SETDB1 and SETDB2 in both BCBL-1 and BC-3 cells (*Figure 3A*, right panel, lanes 7–9). The lack of IP in BJAB cells confirmed the specificity of the signal. However, we did not find an interaction between LANA1 and SUV39H1 in PEL cells. All the other H3K9 MTases also failed to co-IP. ASC, H3, H3K9me3, and HP1α also did not interact with LANA1. In contrast, consistent with Toth et al.'s findings (*Toth et al., 2016*), we observed that LANA1 interacts with EZH2 in BCBL-1 and BC-3 cells. All the corresponding inputs are shown in the middle panel (*Figure 3A*, lanes 4–6).

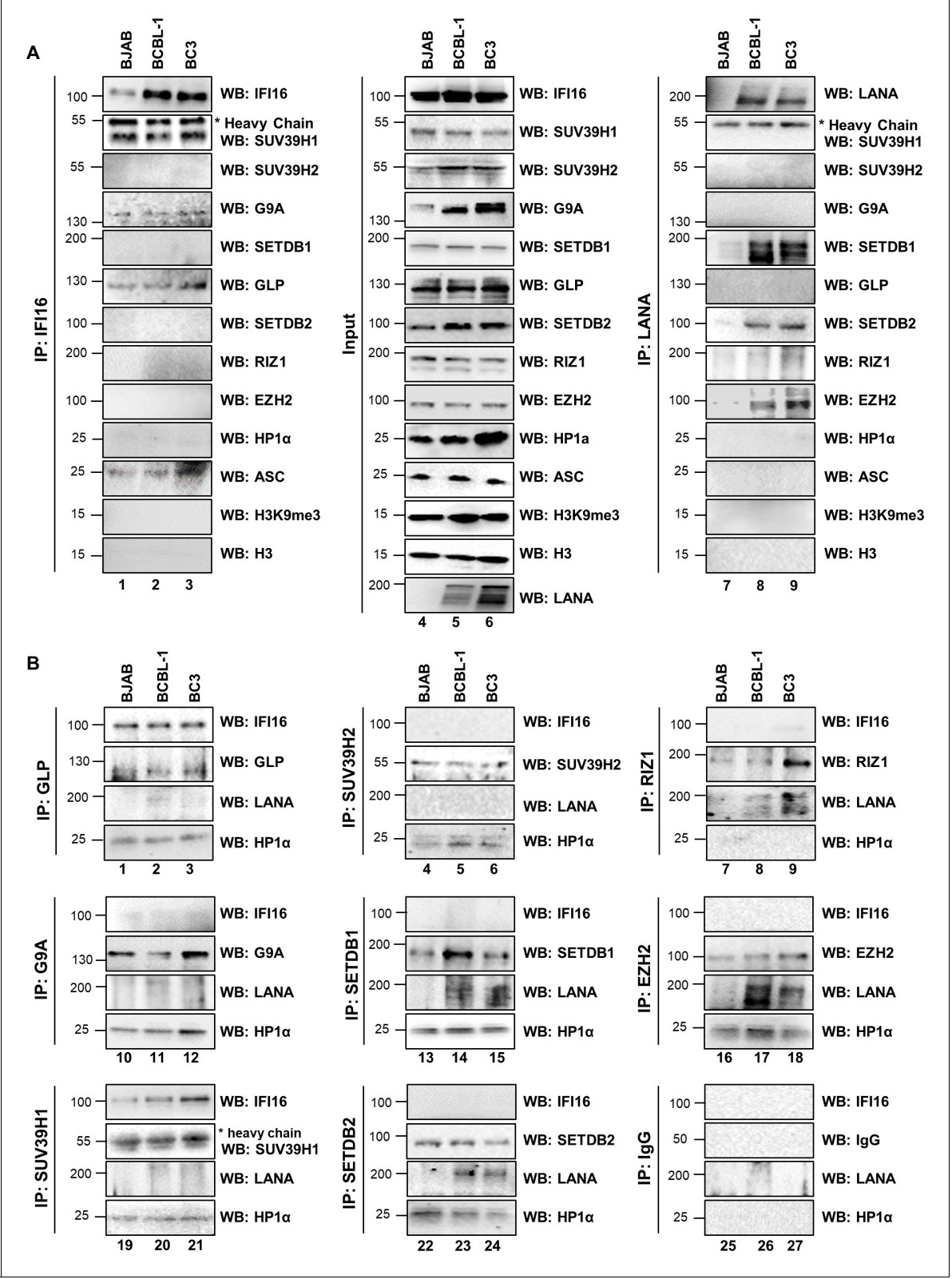

**Figure 3.** Demonstration of IFI16's interaction with specific H3K9 MTases in KSHV latently infected PEL (BCBL-1 and BC-3) cells and in uninfected control BJAB cells. (**A**) Nuclear fractions were isolated from latently infected cells and uninfected BJAB cells and treated with Benzonase. IPs were performed using anti-IFI16 mAb and LANA mAb and WBs were performed. (**B**) To confirm IFI16's and LANA's interaction with H3K9 MTases, IPs were done with Abs against the H3K9 MTases and blotted for the corresponding MTase, IFI16, LANA and HP1α (heterochromatin protein 1α).

DOI: https://doi.org/10.7554/eLife.49500.004

We validated these findings by conducting reverse co-IP experiments where we pulled down the respective H3K9 MTases and blotted for IFI16, LANA, HP1α and the MTase (*Figure 3B*). The observed results corroborated with *Figure 3A* results, and thus confirmed that IFI16 interacts with SUV39H1 and GLP, while LANA1 interacts with SETDB1, SETDB2, and EZH2. However, IP of G9A did not pull-down IFI16. With the exception of RIZ1, all of the H3K9 MTases tested along with EZH2, interacted with HP1α.

## IFI16 interacts with H3K9 MTase SUV39H1 and GLP during de novo KSHV infection

During de novo infection, the linear histone-free input herpes viral DNA is circularized in the nucleus, rapidly adopts a chromatin structure, and viral and host epigenetic factors drive a spatially and temporally ordered recruitment of epigenetic marks onto the viral genome leading to selective silencing of lytic gene expression, while allowing expression of the latent genes (*Toth et al., 2013b*; *Renne et al., 1996*; *Toth et al., 2016*). To decipher IFI16's interactions with H3K9MTases during de novo infection, TIME cells were mock infected or infected with KSHV for 6 or 24 hr, nuclear fractions treated with benzonase and IP-ed with anti-IFI16 antibodies. Confirming our findings in PEL cells (*Figure 3*), we observed that SUV39H1 and GLP were efficiently pulled-down by IFI, while G9A showed weak interactions (*Figure 4A*, left panel). ASC as a positive control interacted with IFI16 while all the other H3K9 MTases, H3, H3K9me3, and HP1α did not interact with IFI16 (*Figure 4A*). All the corresponding inputs are shown in the right panel (*Figure 4A*, lanes 4–6). The reverse co-IP experiments using anti-MTase antibodies and IFI16 WBs also substantiated these observations (*Figure 4B*).

## Ectopically expressed IFI16 interacts with SUV39H1 and GLP in 293T cells

To confirm that the observed interactions between IFI16 and SUV39H1/GLP were not due to non-specific interactions of anti-IFI16 antibody, we ectopically expressed His-tagged full-length IFI16 in 293T cells which lack the expression of endogenous IFI16. His-tag IFII6 co-elution assays also pulled down SUV39H1, GLP and G9A and not the other H3K9MTases (*Figure 4C*). Next, we expressed the untagged IFI16 and LANA1 in 293T cells and IP-ed them with their respective antibodies. Ectopically expressed IFI16 and LANA-1 exhibited similar interactions as observed in PEL cells (*Figure 3*) namely IFI16 interacts with SUV39H1 and GLP, and LANA1 fails to interact with SUV39H1 but interacts with EZH2 (*Figure 4D*). These observations in 293T cells proved that IFI16 interactions with SUV39H1 and GLP H3K9 MTases are specific and not due to non-specific pull-downs.

## IFI16 interacts with H3K9MTase SUV39H1 and GLP and recruits them on to the KSHV genome

To investigate IFI16's ability to recruit these H3K9MTases onto the incoming KSHV genome during de novo infection, we performed EdU-genome pulldown assay after siRNA-mediated KD of IFI16 in TIME cells. 24 hr post-KSHV infection, we observed that KD of IFI16 resulted in reduced recruitment of SUV39H1, GLP, and G9A, but not the other H3K9MTases tested (*Figure 5A*). In addition, confirming our ChIP results in *Figure 1F*, we observed reduced recruitment of H3K9me3 under IFI16 KD conditions. Furthermore, IFI16 KD also resulted in the reduced recruitment of H3K9me2 but not H3K9me1 to the KSHV genome, while the total H3 and H3K27me3 recruitment were unaltered. ChIP experiments in KSHV latently infected BCBL-1 cells confirmed that KD of IFI16 results in reduced recruitment of SUV39H1, GLP and G9A at the KSHV lytic ORFs 63, 25 and 64 promoters (*Figure 5B*, black arrows). These three promoters were chosen since they fall within the KSHV genome regions where H3K9me3 was found to be highly enriched (*Günther et al., 2014*; *Toth et al., 2010*).

To visualize the tripartite interaction between IFI16, H3K9 MTase and the KSHV genome in situ, we performed Proximity Ligation Assay (PLA) in TIME cells 24 hr after infection with EdU-labeled KSHV. PLA is a powerful technique which produces a fluorescent signal if the two interacting proteins are within close proximity of ~40 nm or less. Therefore, only physical interaction to IFI16 will result in a positive PLA signal as opposed to IP-based techniques where interacting partners in a multi-protein complex can also be pulled-down. We observed that both SUV39H1 and GLP produced positive PLA dots/spots (*Figure 5C*, green) with IFI16 confirming their physical interaction. In

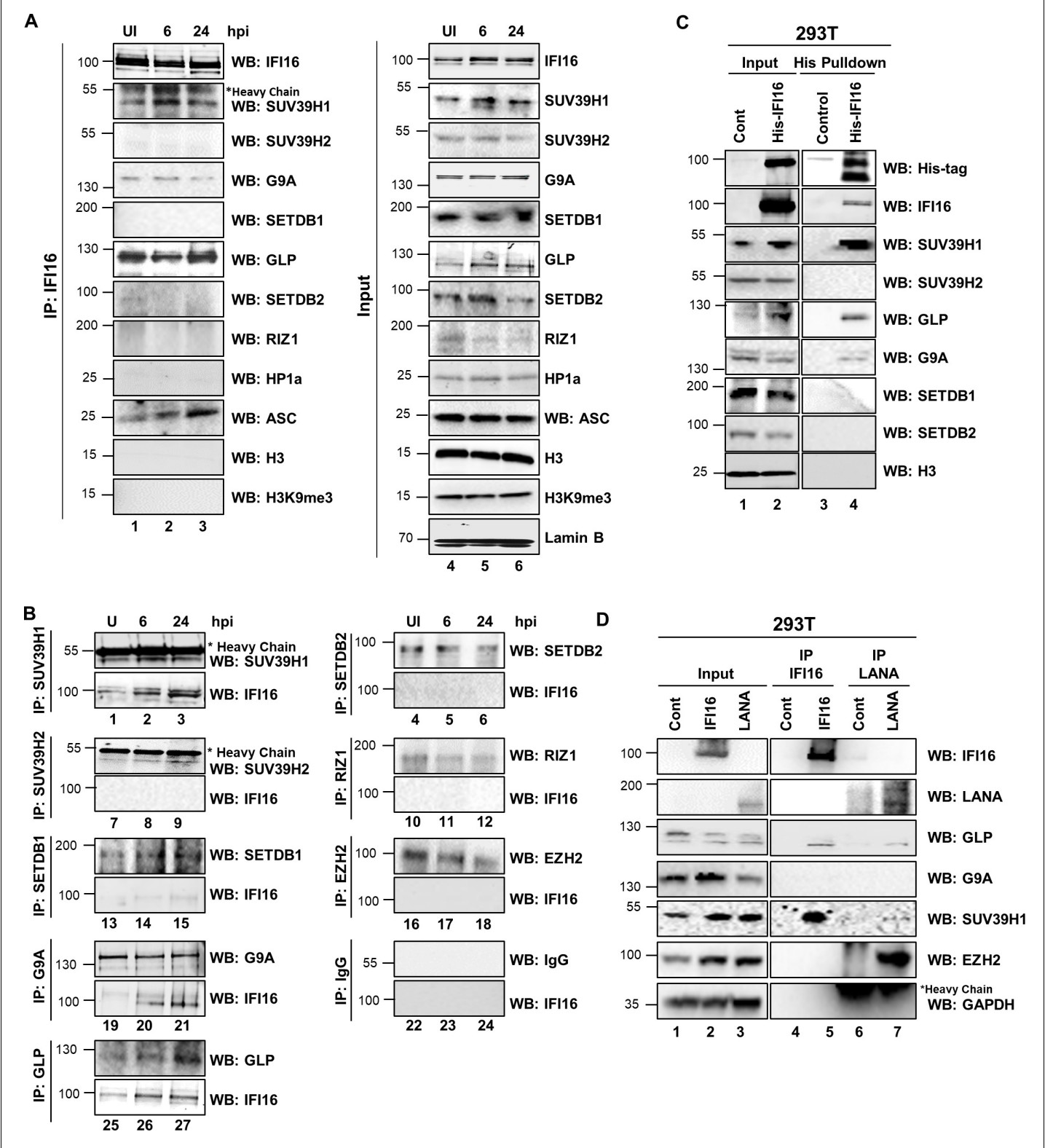

**Figure 4.** Demonstration of IFI16's interaction and recruitment of specific H3K9 MTases during de novo KSHV infection. (**A**) TIME cells either left uninfected or infected with KSHV for 6 or 24 hr were IPed with anti-IFI16 antibodies and western blotted for the indicated proteins. (**B**) To confirm IFI16's interaction with H3K9 MTases, TIME cells were infected as in (A) and IPed with antibodies against the MTases and blotted for the corresponding MTase and IFI16. (**C**) 293 T cells lacking IFI16 transfected with control plasmid or His-IFI16 expressing plasmid for 72 hr were utilized for His-tag pulldown using HisPur cobalt resin. Inputs and elutions were blotted for the indicated proteins. (**D**) 293 T cells transfected with control plasmid, IFI16

*Figure 4 continued on next page*

*Figure 4 continued*

expressing plasmid or LANA expressing plasmid for 72 hr were IPed with anti-IFI16 mAb or LANA mAb. Inputs and elutions were blotted for the indicated proteins.

DOI: https://doi.org/10.7554/eLife.49500.005

addition to PLA, these cells were also stained for the EdU-labeled genome using Click chemistry (*Figure 5C*, red). Colocalization of the green PLA dots of IFI16+GLP and IFI16+SUV39H1 PLA with the red EdU-genome (*Figure 5C*, yellow indicated by white arrows) confirmed that this is a tripartite complex between the genome, IFI16 and the respective H3K9MTase. This interaction of IFI16 with GLP and SUV39H1 was also observed in the uninfected cells. Interestingly, we failed to find PLA signal between G9A and IFI16. Although we detected G9A in IFI16 pull-down experiments (*Figures 3* and *4*), failure to detect it in PLA experiments suggest that G9A may not be directly interacting with IFI16 and may have been pulled-down by virtue of its interaction with GLP that has been shown before (*Shinkai and Tachibana, 2011*). In addition, consistent with previous observations, neither SETDB1 nor the control IgG showed any interaction with IFI16 (*Figure 5C*). Moreover, an IgG + IgG control PLA (*Figure 5—figure supplement 1*) also confirm the specificity of the observed PLA signals.

To further confirm these results, we used IFI16 CRISPR knockout (KO) osteosarcoma U2OS cells (U2OS 67) generated in our earlier studies (*Johnson et al., 2014*) to study the effect of IFI16 KO and re-introduction of IFI16 in a KO background on the recruitment of H3K9me3, SUV39H1 and GLP. U2OS wt and U2OS 67 cells were infected with KSHV for 24 hr, and ChIPs were performed with anti-H3, H3K9me3, SUV39H1 and GLP antibodies. KSHV late ORF63 promoter occupancy was assessed by real-time PCR. Consistent with previous results, we observed reduced recruitment of H3K9me3, SUV39H1 and GLP on the ORF63 promoter in IFI16 KO cells (*Figure 5D*). Total H3 did not decrease under similar conditions. The WB confirms KO of IFI16 (*Figure 5D*). Next, we reintroduced IFI16 (pCDNA3.1+ IFI16) or control GFP (pCDNA3.1+ GFP) in the U2OS 67 KO cells and performed WBs which confirmed the expression of IFI16 and GFP (*Figure 5E*). When we performed the same experiment described in *Figure 5B*, we observed that rescue with IFI16 resulted in about threefold increase in H3K9me3 recruitment and about 1.5-fold increase in SUV39H1 and GLP recruitment on the KSHV late ORF63 promoter in U2OS 67 cells (*Figure 5E*). Together, these results suggested that IFI16 is instrumental in recruiting H3K9MTase SUV39H1 and GLP onto KSHV lytic promoters during de novo infection, resulting in deposition of the heterochromatin H3K9me3 mark.

**PLA in IFI16 KO cells confirms that IFI16 recruits SUV39H1 and GLP onto the KSHV genome resulting in the deposition of H3K9me3 but not H3K27me3**

To confirm the IFI16-mediated recruitment of heterochromatic factors at a single-cell level, we infected U2OS wt and IFI16 KO U2OS67 cells with BrdU-labeled KSHV and preformed PLA for BrdU and the indicated proteins. The expectation for this was that positive PLA signal using antibody against BrdU and antibodies against the proteins of interest will visually confirm the physical proximity/interaction between the BrdU-labeled KSHV genome and the protein of interest. In addition to PLA (*Figure 6*, red), we also immunostained for IFI16 (*Figure 6*, green). We observed numerous BrdU genome + H3K9me3 PLA dots in the U2OS wt cells confirming efficient recruitment of H3K9me3 on the KSHV genome in these cells (*Figure 6A*, red dots, white arrows). However, the number of PLA dots in U2OS 67 cells were significantly reduced (*Figure 6A*, blue arrow) confirming the role of IFI16 in the recruitment of H3K9me3. In contrast, we did not observe any reduction in the BrdU genome + H3K27me3 PLA dots in the U2OS 67 cells compared to the wt type cells (*Figure 6B*, red dots). Similar experiments for BrdU genome + SUV39H1 PLA and BrdU genome + GLP PLA demonstrated significantly reduced PLA dots in the IFI16 KO U2OS 67 cells compared to the wt type cells (*Figure 6C and D*). The absence of significant PLA dots in the control BrdU+IgG PLA (*Figure 6—figure supplement 1*) confirmed the specificity of the PLA reaction. Consistent with previous reports (*Ansari et al., 2015*), we observed IFI16's redistribution to the cytoplasm after KSHV infection in U2OS wt cells.

Next, we performed similar experiments in U2OS 67 cells transfected with either control (pcDNA3.1+) plasmid or IFI16 expressing plasmid (pcDNA3.1+ IFI16) for 72 hr. We included uninfected (UI) condition to assess the specificity of the PLA reactions (*Figure 7—figure supplement 1*).

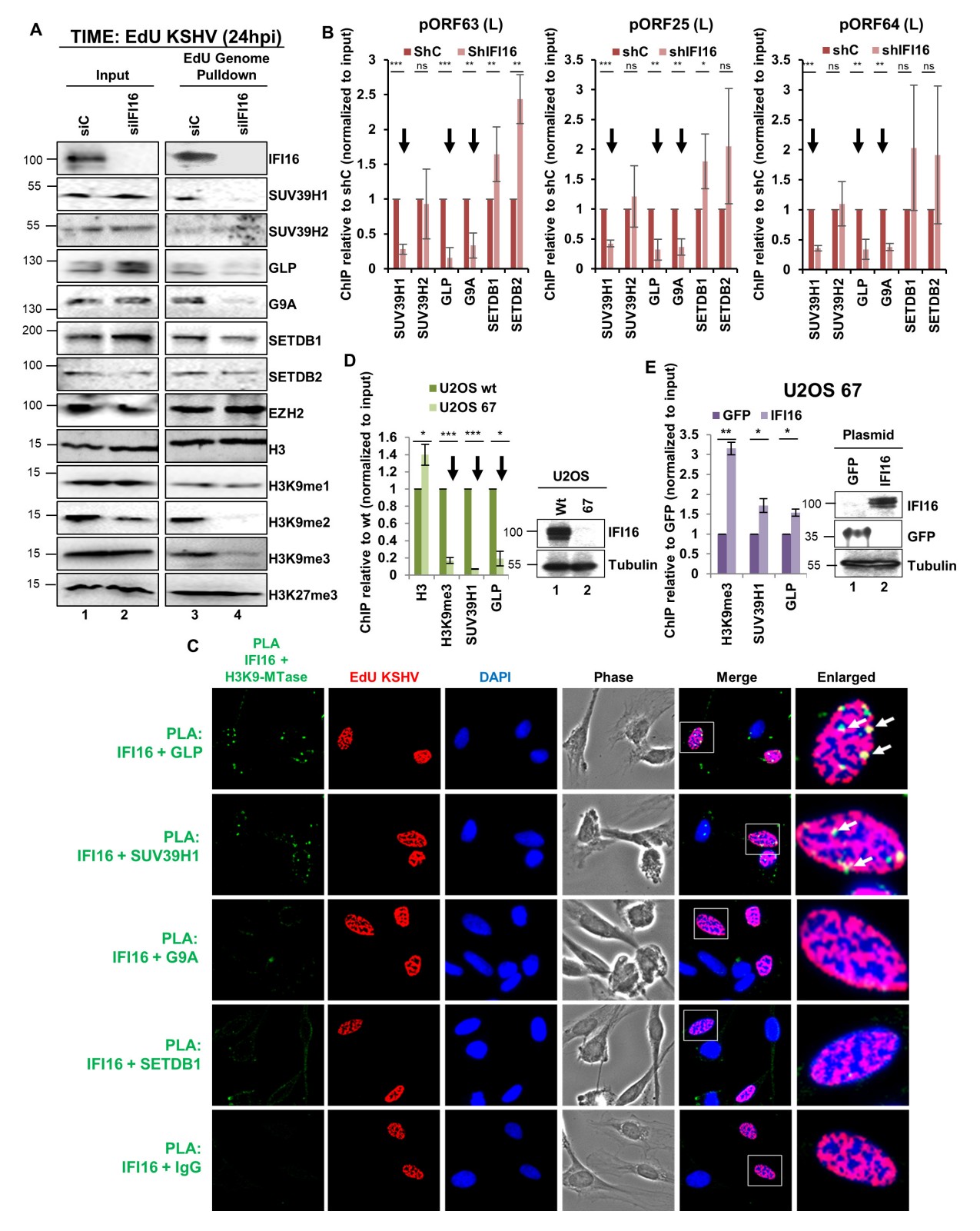

**Figure 5.** Demonstration of IFI16's specific interaction with GLP and SUV39H1 and the effect of IFI16 KD upon the recruitment of SUV39H1 and GLP to the KSHV genome during de novo infection. (**A**) IFI16 was KD in TIME cells using siRNA and 72 hr later, cells were infected with EdU-KSHV for 24 hr followed by EdU-KSHV genome pulldown using Click chemistry. The inputs and eluates were blotted for different H3K9 MTases and H3K9 methylations. (**B**) ChIP was performed after shRNA IFI16 KD in BCBL-1 cells and recruitment of different H3K9 MTases on different KSHV promoters (pORF63- L,
*Figure 5 continued on next page*

*Figure 5 continued*

pORF25- L and pORF64- L) representing promoters where H3K9me3 is most abundantly recruited, were tested by q-PCR. ChIP efficiencies were normalized to input chromatin and are represented as relative to shC control. (C) TIME cells were infected with EdU-KSHV for 24 hr and stained using the Click-iT EdU Alexa Fluor 594 Imaging Kit (red). Subsequently, Proximity Ligation Assay (PLA) was performed to assess the interaction between IFI16 and the indicated H3K9 MTases (green). Colocalization of green (PLA) with red (EdU-KSHV genome) resulting in yellow indicates interaction of IFI16 with the corresponding H3K9 MTases on the KSHV genome (enlarged image, white arrows). Uncropped source PLA data for *Figure 5C* showing a larger field containing multiple cells are shown in *Figure 5—figure supplements 2* and *3*. (D) U2OS wt and U2OS 67 (CRISPER Cas-9 IFI16 KO) cells were infected with KSHV (100 DNA copies/cell) for 24 hr, ChIP performed, and the late KSHV promoter pORF63 tested by q-PCR. ChIP efficiencies were normalized to input chromatin and are represented as relative to U2OS wt. WB shows IFI16 KO in U2OS 67 cells. (E) U2OS 67 cells were transfected with either control GFP plasmid or IFI16 plasmid. After 72 hr, cells were infected with KSHV for 24 hr, ChIP performed, and the late KSHV ORF63 promoter was tested by q-PCR. ChIP efficiencies were normalized to input chromatin and are represented as relative to GFP transfected control. The WB shows IFI16 and GFP overexpression in U2OS 67 cells following transfection. Data shown are averages of the results of at least three experiments ± SD (*, p<0.05; **, p<0.01; ***, p<0.001).

DOI: https://doi.org/10.7554/eLife.49500.006

The following figure supplements are available for figure 5:

**Figure supplement 1.** Assessment of the specificity of the PLA reaction.

DOI: https://doi.org/10.7554/eLife.49500.007

**Figure supplement 2.** Uncropped source PLA data for *Figure 5C*.

DOI: https://doi.org/10.7554/eLife.49500.008

**Figure supplement 3.** Uncropped source PLA data for *Figure 5C* (continued).

DOI: https://doi.org/10.7554/eLife.49500.009

---

Upon BrdU-KSHV infection of the U2OS 67 cells transfected with the control plasmid, very few PLA dots were observed between the BrdU-genome and H3K9me3 (*Figure 7A*, first row panels, white arrows). However, upon rescue with IFI16 plasmid transfection, the number of PLA dots increased significantly in the cells expressing IFI16 (*Figure 7A*, second row panels, white arrows), but not in cells in which IFI16 expression was minimal or absent (*Figure 7A*, orange arrow). Similar observations were also made with SUV39H1 (*Figure 7B*) and GLP (*Figure 7C*). The absence of significant PLA dots in the UI samples (*Figure 7—figure supplement 1*) and in the control BrdU+IgG PLA (*Figure 6—figure supplement 1*) confirmed the specificity of the PLA reaction. Together, these observations confirmed that IFI16 is responsible for the recruitment of SUV39H1 and GLP on the KSHV genome eventually leading to the addition of the H3K9me3 histone mark.

## H3K9 MTase SUV39H1 and GLP are essential for KSHV gene regulation but not G9A

Since our results have demonstrated that IFI16 binds and recruits SUV39H1 and GLP onto the KSHV genome, we examined the dynamics of these two H3K9MTases and IFI16 on the KSHV genome during lytic reactivation of TREXBCBL-1 cells. We observed that the abundance of SUV39H1 on the KSHV genome decreased to about 25% of that of uninduced on day 2 post-induction and remained at that level until day 4. In contrast, a more drastic decrease was observed with GLP which was reduced to less than 10% of the uninduced level by day 1 post-induction. Abundance of IFI16 decreased more gradually and reduced to less than 20% by day 4, which is also probably due to the degradation of IFI16 during lytic reactivation (*Roy et al., 2016*). Next, we investigated the role of these two H3K9MTases in KSHV life cycle, and included G9A as well in this assay as it is known to exist in a complex with GLP (*Shinkai and Tachibana, 2011*). These three MTases were knocked down in BCBL-1 cells via lentivirus-mediated shRNA for 48 and 96 hr and the expression of all four temporal KSHV gene classes were measured by real-time qRT-PCR. KD efficiencies for SUV39H1, GLP and G9A were tested by WB (*Figure 8B,D and F*, respectively). 48 hr post SUV39H1 KD, KSHV gene expression for all lytic genes tested (IE, E, and L) increased significantly and remained elevated at 96 hr post-KD (*Figure 8C*). However, latent genes increased only marginally at 48 hr while at 96 hr, no increase was observed. KD of GLP had more dramatic outcomes at 96 hr post IFI16 KD. In this case, lytic gene expressions were induced between 25 and 45-fold (*Figure 8E*). Here too, latent gene expression increased only marginally. However, at 48 hr post-GLP KD, lytic genes were not induced significantly. In contrast to SUV39H1 and GLP, we observed that G9A KD failed to induce lytic gene expressions at both time points (*Figure 8G*). These results suggested that: a) H3K9MTase

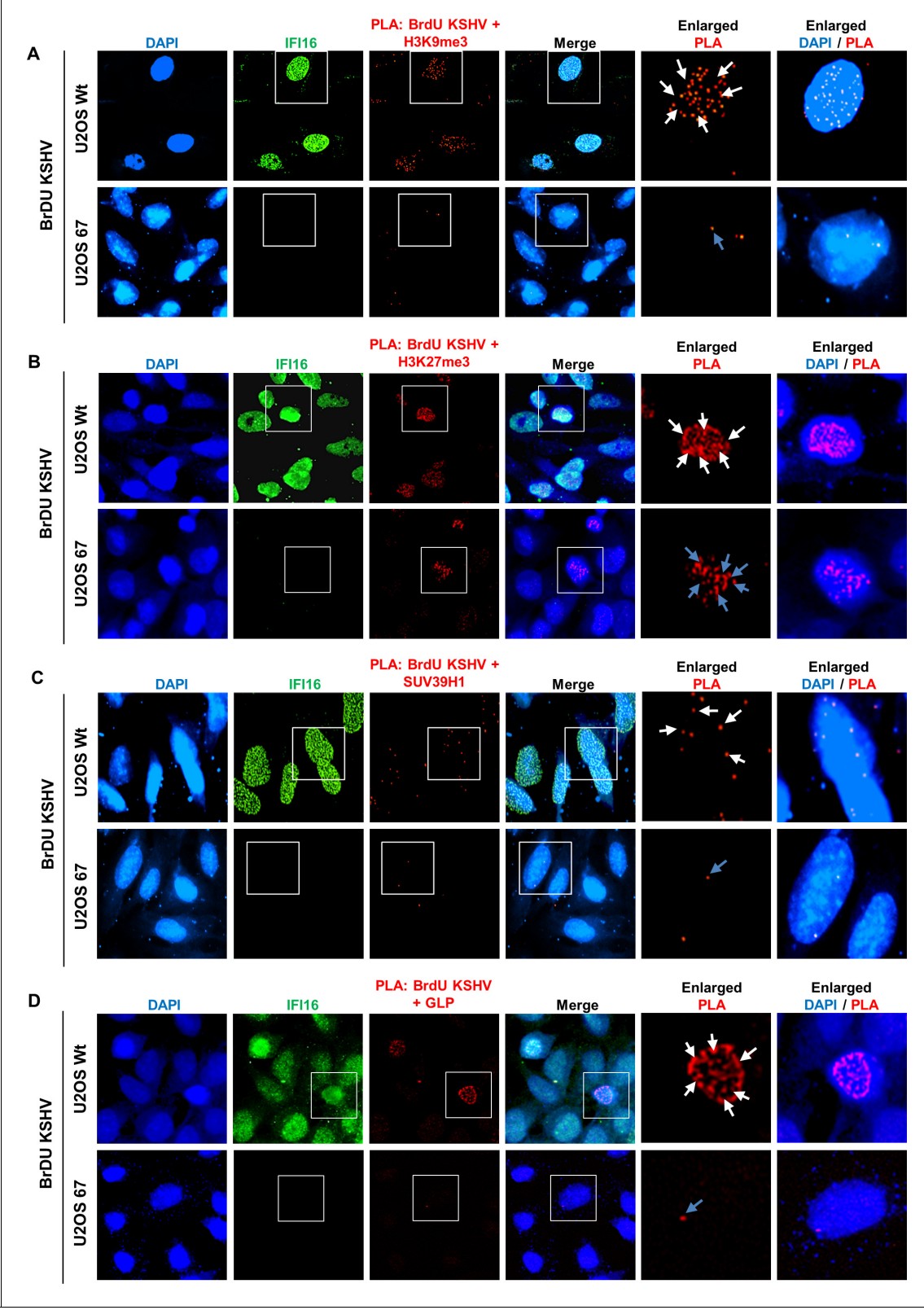

**Figure 6.** Proximity Ligation Assay (PLA) demonstrating the reduced recruitment of H3K9me3, GLP and SUV39H1 but not H3K27me3 onto the KSHV genome by IFI16 KD. (**A**) U2OS wt and U2OS 67 (IFI16 KO) were infected with BrdU genome labeled KSHV for 24 hr and PLA was performed to assess the association between BrdU-KSHV genome DNA and H3K9me3 (red). Following PLA, IFA was performed to stain for IFI16 (green). Colocalization of green (IFA) with red (PLA) resulting in yellow indicates the presence of both, IFI16 and H3K9me3 on the KSHV genome (merged image). In the U2OS 67

*Figure 6 continued on next page*

*Figure 6 continued*

panel, as there is no expression of IFI16, yellow colocalization is absent. Boxed areas are enlarged. The number of PLA dots (red) were compared between the U2OS wt (white arrows) and 67 panels (blue arrows). Similar experiments were performed for H3K27me3 (B), GLP(C), and SUV39H1 (D). Uncropped source PLA data for *Figure 6* showing a larger field containing multiple cells are shown in *Figure 6—figure supplements 2* and *3*.

DOI: https://doi.org/10.7554/eLife.49500.010

The following figure supplements are available for figure 6:

**Figure supplement 1.** Assessment of the specificity of the PLA reaction.

DOI: https://doi.org/10.7554/eLife.49500.011

**Figure supplement 2.** Uncropped source PLA data for *Figure 6A and B*.

DOI: https://doi.org/10.7554/eLife.49500.012

**Figure supplement 3.** Uncropped source PLA data for *Figure 6C and D*.

DOI: https://doi.org/10.7554/eLife.49500.013

SUV39H1 and GLP are important factors for the maintenance of KSHV latency and their depletion induces lytic reactivation, and b) G9A, although in a complex with GLP, has a minimal role in KSHV transcriptional regulations.

We also investigated whether depletion of SUV39H1, GLP and G9A can result in KSHV genome replication and lytic reactivation. For this, we determined the KSHV genome DNA copy numbers in BCBL-1 cells treated with shRNAs against these 3 MTases. We observed that only GLP KD resulted in a two fold increase in the intracellular viral genome copy numbers after 4 days of shRNA treatment (*Figure 8H*).

Next, we examined the effect of SUV39H1 and GLP KD during de novo infection of TIME cells. TIME cells were electroporated with the corresponding siRNA and 72 hr later, infected with KSHV for 48 hr. Efficient KD of SUV39H1 and GLP were confirmed by WB (*Figure 8I and K*, respectively). We observed that KD of SUV39H1 and GLP MTases results in an increase in the lytic ORF50 transcripts, while latent ORF73 transcripts remained predominantly unaffected (*Figure 8J and K*, respectively). These observations supported the results observed in PEL cells and confirmed the importance of SUV39H1 and GLP MTases in the establishment of KSHV latency.

## Recruitment of heterochromatin protein 1α (HP1α) on the KSHV genome is dependent on IFI16 mediated H3K9-trimethylation

In eukaryotes, chromatin marked by H3K9me2/me3 serves as a binding site for HP1α, a chromodomain containing heterochromatin protein. Upon binding to the chromatin, HP1α self-oligomerizes and recruits multiple repressive histone modifiers, which ultimately leads to heterochromatin compaction and spread (*Eissenberg and Elgin, 2014*). We therefore asked whether IFI16-mediated addition of H3K9me2/me3 marks on the KSHV genome results in the recruitment of HP1α on to the viral genome. We first tested if HP1α is recruited onto the KSHV genome during de novo infection. Immunostaining for HP1α (*Figure 9A*, green) in TIME cells infected with EdU-labeled KSHV showed that HP1α colocalizes (*Figure 9A*, yellow) with the EdU genome (*Figure 9A*, red) at 24 and 48 h p.i. (*Figure 9A*, white arrows).

To determine if HP1α-mediated heterochromatin formation is essential for KSHV gene regulations, we KD HP1α via lentivirus mediate shRNA in BCBL-1 cells. 96 hr post-KD, efficiency of the KD was assessed by real-time qRT-PCR (*Figure 9B*) and WB (*Figure 9C*). Results showed that the shRNA pool specifically KD HP1α and not HP1β and HP1γ, the other two isoforms of HP1α (*Figure 9B*). When we measured the expression of all four temporal KSHV gene classes by real-time qRT-PCR (*Figure 9D*), we observed that HP1α KD results in the substantial increase in the expression of all lytic genes (IE, E, and L) with no significant change in the latent gene expression. Expression of IFI16 mRNA which was used as a host gene control did not change significantly. These results confirmed that HP1α is an important factor for the KSHV heterochromatic gene regulations.

To investigate the role of IFI16 in the recruitment of HP1α, we infected TIME cells with EdU-KSHV for 24 hr. Pulldown of the EdU-genome yielded HP1α which suggested that HP1α recruitment on the KSHV genome can be studied by the EdU-genome pulldown assay (*Figure 9E*). In a similar assay when we KD IFI16 using siRNA, we observed a significant reduction in the KSHV genome associated HP1α pulldown (*Figure 9F*) which confirmed the role of IFI16 in recruiting HP1α (*Figure 9F*). In addition, we also performed ChIP for HP1α in BCBL-1 cells and observed that KD of IFI16 results in

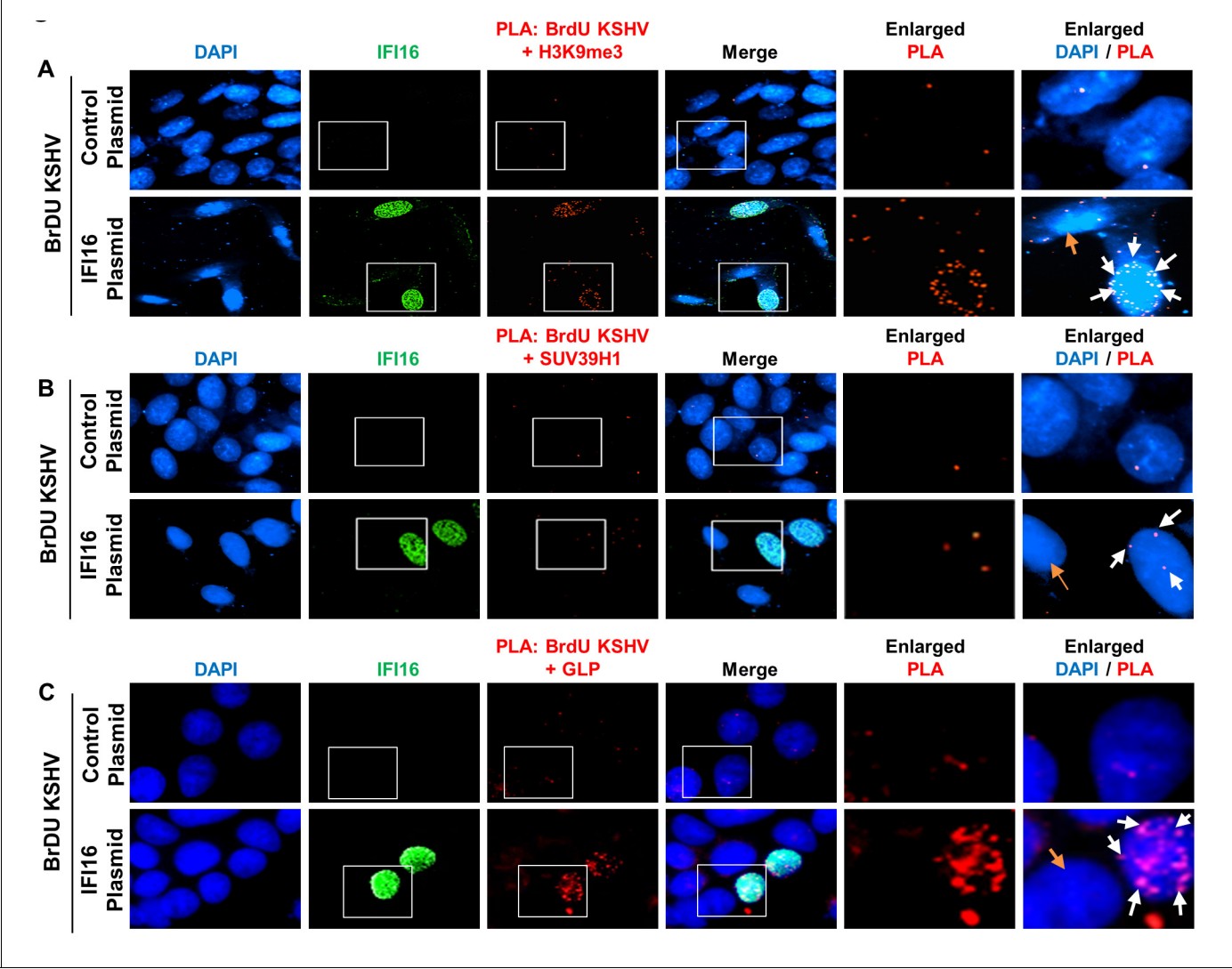

**Figure 7.** Demonstration of increased recruitment of H3K9me3, GLP and SUV39H1 onto the KSHV genome after IFI16 rescue of IFI16 KO U2OS 67 cells. (**A**) U2OS 67 cells were transfected with control plasmid or IFI16 expressing plasmid. After 72 hr, cells were infected with BrdU-KSHV for 24 hr. Following infection, PLA was performed to assess the interaction between the BrdU-KSHV genome DNA and H3K9me3 (red). Subsequently, IFA was performed to stain for IFI16 (green). The number of PLA dots (red) can be compared between the different panels. Boxed areas are enlarged. The white arrows show PLA dots in a cell expressing transfected IFI16, while the orange arrow show a cell that has not been transfected with IFI16 in the same field. A similar experiment was performed for SUV39H1 (**B**) and GLP (**C**). Uncropped source PLA data for **Figure 7** showing a larger field containing multiple cells are shown in **Figure 7—figure supplements 2** and **3**.

DOI: https://doi.org/10.7554/eLife.49500.014

The following figure supplements are available for figure 7:

**Figure supplement 1.** Assessment of the specificity of the PLA reaction.
DOI: https://doi.org/10.7554/eLife.49500.015

**Figure supplement 2.** Uncropped source PLA data for **Figure 7A and B**.
DOI: https://doi.org/10.7554/eLife.49500.016

**Figure supplement 3.** Uncropped source PLA data for **Figure 7C**.
DOI: https://doi.org/10.7554/eLife.49500.017

significantly reduced recruitment of HP1α on the different KSHV promoters (**Figure 9G**). As we observed in **Figure 3A** that IFI16 does not directly interact with HP1α, this suggested that HP1α recruitment dependency on IFI16 is due to IFI16's ability to establish H3K9me3 marks on the KSHV genome which serves as a docking site for HP1α.

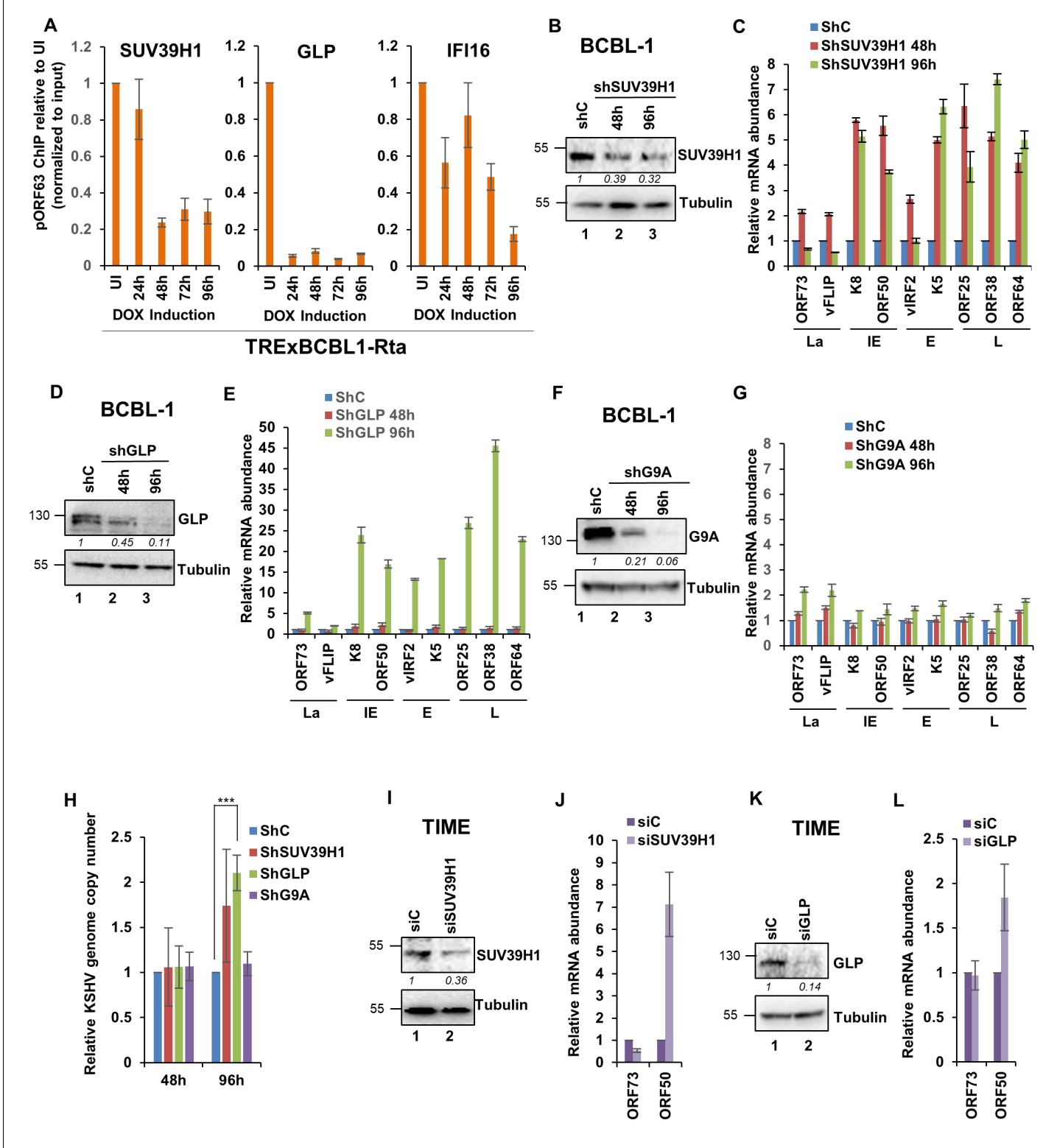

**Figure 8.** Demonstration of the essential role of H3K9MTAses SUV39H1, and GLP but not G9A in the regulation of KSHV genes during latency and de novo infection. (A) TRExBCBL1-RTA cells were induced with doxycycline and at 0, 1, 2, 3 and 4 days post-induction, ChIP was performed against the indicated proteins. Deposition of GLP, SUV39H1 and IFI16 on the ORF63 promoter was tested by q-PCR. ChIP efficiencies normalized to input chromatin are shown as relative to uninduced (UI) control. (B) SUV39H1 was knocked down in BCBL-1 cells using lentivirus shRNA for 72 hr and KD efficiency was assessed by WB. (C) mRNA levels of KSHV genes were assessed by real-time RT-PCR after SUV39H1 KD. (D) GLP was knocked down in

*Figure 8 continued on next page*

*Figure 8 continued*

BCBL-1 cells using lentivirus shRNA for 72 hr and KD efficiency assessed by WB. (**E**) mRNA levels of KSHV genes were assessed by real-time RT-PCR after GLP KD. (**F**) G9A was knocked down in BCBL-1 cells using lentivirus shRNA for 72 hr and KD efficiency assessed by WB. (**G**) mRNA levels of KSHV genes were assessed by real-time RT-PCR after G9A KD. All mRNA levels were normalized against β-tubulin mRNA and are expressed as relative amounts compared to shC treatments. (**I**) TIME cells were electroporated with siC or a SUV39H1-specific siRNA pool. After 72 hr, cells were de novo infected with KSHV DNA copies/cell) for 48 hr. SUV39H1 KD efficiency was assessed by WB. (**H**) Real-time DNA-PCR showing the KSHV genome copy numbers at 2 and 4 days post-KD of SUV39H1, GLP and G9A in BCBL-1 cells. Primers specific to the ORF73 gene were used and the level of genomic DNA was normalized against the β-tubulin gene. (**J**) mRNA expression of ORF73 and ORF50 genes was evaluated using the TaqMan method, normalized to cellular RNaseP and are expressed as relative to siC treatment. (**K**) GLP KD efficiency was assessed by WB in TIME cells that were electroporated with siC or a GLP-specific siRNA pool. After 72 hr, cells were de novo infected with KSHV for 48 hr. (**L**) mRNA expression of ORF73 and ORF50 genes was evaluated by TaqMan method, normalized to cellular RNaseP and are expressed as relative to siC treatment. Data shown are averages of the results of at least two experiments ± SD *, p<0.05; **, p<0.01; ***, p<0.001.
DOI: https://doi.org/10.7554/eLife.49500.018

Next, to ascertain the relevance of HP1α in the latency to lytic transition of KSHV, we studied the dynamics of its recruitment on the KSHV genome in doxycycline-induced TREXBCBL-1 cells by ChIP at different times post-induction (*Figure 9H*). HP1α abundance on the KSHV genome significantly reduced by about 60% during reactivation suggesting its active role in KSHV latency. We also investigated whether depletion of HP1α resulted in KSHV genome replication and lytic reactivation in BCBL-1 cells, and observed that KD of HP1α fold did not significantly induce the KSHV genome replication (*Figure 9I*).

## Discussion

The transcriptional repressor function of IFI16 has been identified in diverse experimental models (*Caposio et al., 2007*; *Johnstone et al., 1998*; *Kang et al., 2014*; *Thompson et al., 2014*; *Roy et al., 2016*). In addition, IFI16's ability to restrict transcription and replication of episomal viral DNA of KSHV, HSV-1, HCMV and HPV18 is also well established (*Roy et al., 2016*; *Gariano et al., 2012*; *Orzalli et al., 2013*; *Merkl and Knipe, 2019*; *Johnson et al., 2014*; *Lo Cigno et al., 2015*). Although these studies suggested that IFI16 may have the ability to promote epigenetic modification of foreign/viral DNA leading to transcriptional silencing (*Orzalli et al., 2013*; *Merkl and Knipe, 2019*; *Johnson et al., 2014*; *Lo Cigno et al., 2015*), till date, no precise epigenetic function of IFI16 has been identified that can explain its ability to inhibit transcription. Our comprehensive studies here demonstrate that under physiological conditions, IFI16 is associated with the H3K9 methyltransferases SUV39H1 and GLP which mediates the deposition of H3K9me2/me3, and thus unravels one of the potential mechanisms by which IFI16 mediates epigenetic modifications and transcriptional regulation.

Lysines 4, 9, 27, 36 and 79 of histone H3, and 20 of histone H4 can be methylated, but, no attempt has been made to comprehensively study the possible role of IFI16 in recruiting these H3 lysine tri-methylations. Hence, we undertook a systamatic approach toward identifying the H3 lysine tri-methylations that can be modulated by IFI16 on the KSHV epigenome and tested all five identified lysine tri-methylations of H3. Our studies demonstrate that depletion of IFI16 results in increased recruitment of H3K4me3 and decreased recruitment of H3K9me3 on the KSHV promoters in both latent BCBL1 cells and in de novo infected TIME cells (*Figure 1C and F*). Both these epigenetic modifications are conducive of increased transcriptional activity which is in agreement with our previous observation that KD of IFI16 results in increased transcription of lytic genes (*Roy et al., 2016*). As a further confirmation of this, we found that KD of IFI16 resulted in increased recruitment of RNA Pol II on all the lytic promoters tested (*Figure 1C and F*). It is interesting to note that *Toth et al. (2010)* and *Günther and Grundhoff (2010)* found that abundant levels of H3K9me3 recruitment was restricted mainly to two regions of KSHV genome (30–60 kb and 95–115 kb) encoding a number of late genes (ORF16–40 and ORF58–68) in TRExBCBL1-Rta cells. However, lesser basal levels were detectable throughout the KSHV genome. Interestingly, *Günther and Grundhoff (2010)* did not observe significant deposition of H3K9me3 on the KSHV genome during de novo infection of SLK cells. However, it must be considered that SLK cells are of epithelial origin and have been shown to be a contaminant from the renal-cell carcinoma cell line Caki-1 (*Stürzl et al., 2013*).

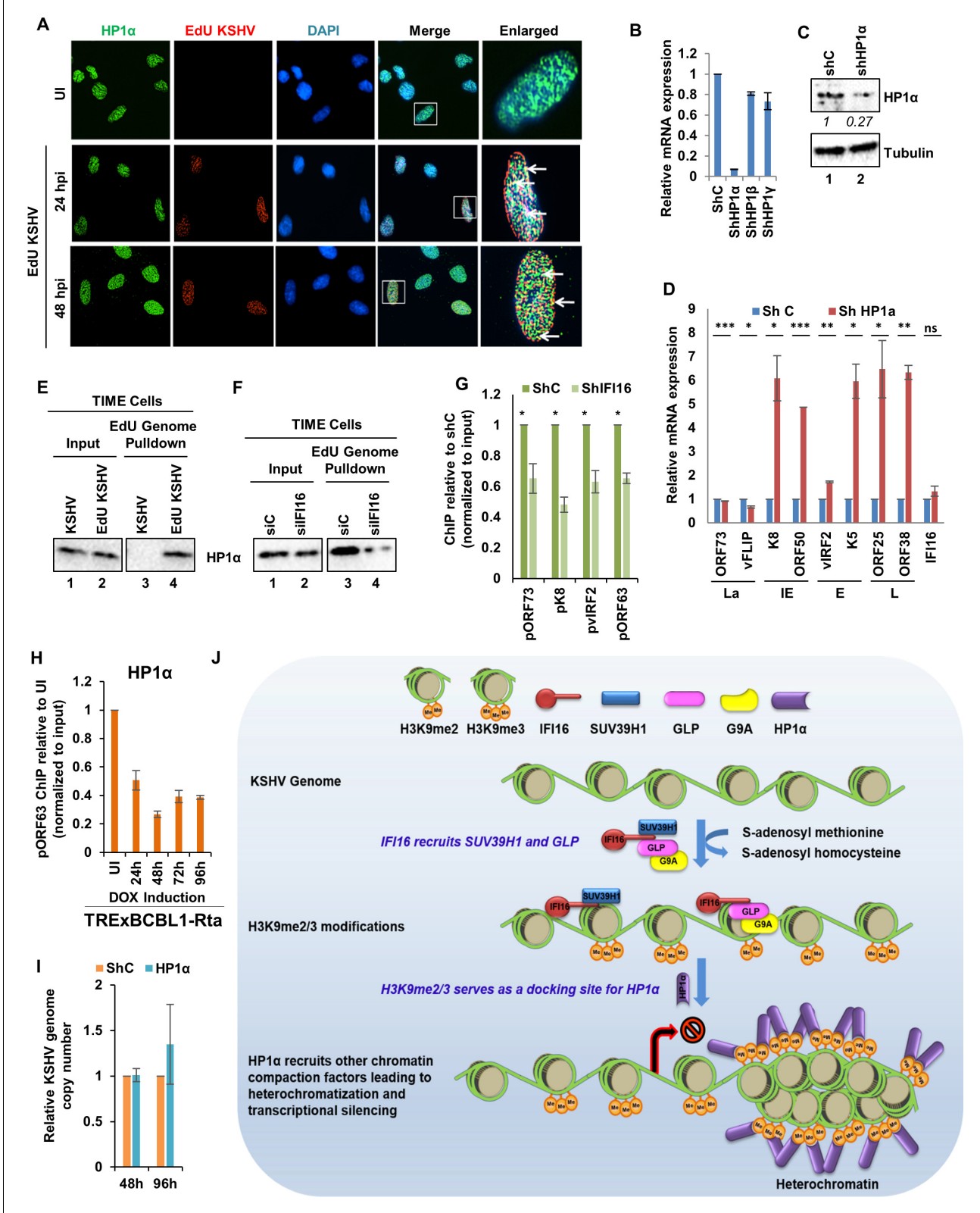

**Figure 9.** Demonstration of IFI16 mediated H3K9me3 dependent recruitment of Heterochromatin Protein 1-α (HP1α). (**A**) TIME cells were infected with EdU-KSHV and stained using the Click-iT EdU Alexa Fluor 594 Imaging Kit (red). Subsequently, IFA was performed against HP1α (green). Colocalization of green (IFA) with red (EdU-KSHV genome) resulting in yellow indicates recruitment of HP1α on the KSHV genome (enlarged image, white arrows). (**B**) HP1α was KD in BCBL-1 cells by shRNA for 96 hr and KD efficiency assessed by qRT PCR using primers specific for HP1α as well as HP1β and HP1γ. (**C**)
*Figure 9 continued on next page*

Figure 9 continued

WB to confirm efficient KD. (D) KSHV mRNA levels were assessed by qRT PCR after HP1α KD. IFI16 mRNA was also assessed to confirm that the effect is specific for the KSHV genome only. mRNA levels were normalized against β-tubulin mRNA and expressed as relative amounts compared to shC-treated cells. (E) TIME cells were either infected with EdU-KSHV or control KSHV for 24 hr followed by EdU-KSHV genome pulldown using Click chemistry. The inputs and eluates were blotted for the presence of HP1α. (F) IFI16 was KD in TIME cells using siIFI16. After 72 hr, cells were infected with EdU-KSHV for 24 hr followed by EdU-KSHV genome pulldown using Click chemistry. The inputs and eluates were blotted for the presence of HP1α. (G) HP1α ChIP was performed 48 hr of de novo infection of TIME cells previously KD of IFI16 for 72 hr. KSHV promoters pORF73- La, pK8- IE, pvIRF2- E, and pORF63- L were tested by q-PCR. ChIP efficiencies have been normalized to input chromatin and are represented as relative to shC control. Data shown are averages of the results of at least three experiments ± SD (*, p<0.05; **, p<0.01; ***, p<0.001. (H) TREXBCBL1-RTA cells were induced with doxycycline and at 0, 1, 2, 3 and 4 days post-induction, ChIP was performed against HP1α. Deposition of HP1α on the ORF63 promoter was tested by q-PCR. ChIP efficiencies normalized to input chromatin are shown as relative to uninduced (UI) control. (I) Real-time DNA PCR showing KSHV genome copy number 2 and 4 days post KD of HP1α in BCBL-1 cells. Primers specific to the ORF73 gene were used and the level of genomic DNA was normalized against the β-tubulin gene. (J) Schematic model showing the role of IFI16 in recruiting and maintaining H3K9 MTase SUV39H1 and GLP onto the KSHV genome leading to tri-methylation (me3) of H3K9. Establishment of H3K9me3 marks on the KSHV genome leads to the recruitment of heterochromatin protein HP1α which in turn leads to the DNA compaction and transcription silencing.
DOI: https://doi.org/10.7554/eLife.49500.019

In contrast, we used the hTERT immortalized human dermal microvascular endothelial cell line, TIME, which we believe is a more suitable cell model for KSHV de novo infection.

After observing that IFI16 regulates the recruitment of H3K9me3, we first asked the question 'Is the H3K9me3 mark important for the KSHV life cycle?". Our studies demonstrate that after lytic induction of TRExBCBL1-Rta cells, levels of H3K9me3 reduced by half on the late ORF63 promoter as early as day 1 (*Figure 1I*). The location of the ORF63 promoter on the KSHV genome is between 102 and 103 kb which is within the high abundance H3K9me3 distribution region described previously (*Toth et al., 2010*; *Günther and Grundhoff, 2010*). In contrast, abundance of H3K27me3 on the ORF63 promoter decreased more gradually between days 2 and 4 post-induction. This strongly suggested that H3K9me3 does play a role in the KSHV lifecycle. Further evidence of the role of H3K9me3 in KSHV lifecycle became evident when we treated BCBL1 cells with A366, a specific inhibitor of H3K9me2 and H3K9me3 methyltransferase (*Kaniskan et al., 2018*). This caused significantly higher transcription of KSHV lytic genes, while expression of the latent genes and the control GAPDH and IFI16 genes did not alter significantly (*Figure 2B*). This is a hallmark of successful lytic reactivation of the latent KSHV genome. Thus, H3K9me2/3 recruitment and maintenance are indispensable for the maintenance of latency. This experiment also suggests that the H3K9me2/3 mark and possibly all the other epigenetic marks on the KSHV genome is dynamic in nature and must be continuously recruited to maintain latency. This may be solely or partly due to the fact that as the BCBL1 cell divides, the newly replicated latent epigenetically naive KSHV genomes must recruit the proper epigenetic marks to be effectively silenced in order to maintain latency.

It is generally considered that H3K9me3 is a hallmark of constitutive heterochromatin while H3K27me3 is typically seen in facultative heterochromatin. Constitutive heterochromatin are often considered to be irreversible such that they will never revert to euchromatin, as those seen at the telomeres and centromeres in mammalian cells. Alternatively, facultative heterochromatin is considered to be reversible and capable of undergoing active gene expression. However, resent advances disprove this notion and H3K9me3's role in cell-type-specific regulation of facultative heterochromatin has been established (*Becker et al., 2016*). Moreover, numerous reports have demonstrated that herpes viral and retroviral genomes are associated with H3K9me3 during latency and almost all these genes undergo active transcription during lytic reactivation (*Günther and Grundhoff, 2010*; *Toth et al., 2010*; *Orzalli et al., 2013*; *Lo Cigno et al., 2015*; *Merkl and Knipe, 2019*; *Bloom et al., 2010*; *Cliffe et al., 2009*).

Our observations confirm that IFI16 recruits the H3K9MTases SUV39H1 and GLP on to the KSHV genome resulting in the enrichment of H3K9me2/me3 marks (*Figures 3–7*). GLP and its partner G9A are known to be responsible for mono and di-methylation (me1/me2) of H3K9 (*Tachibana et al., 2005*). On the other hand, SUV39H1 has been established to use mono and di-methylated H3K9 as a primary substrate and establish tri-methyl H3K9 (*Rea et al., 2000*; *Peters et al., 2001*). Therefore, we believe, that the establishment of H3K9me3 heterochromatic locus on the KSHV genome requires the concerted action of GLP and SUV39H1. The fact that IFI16 interacts with and recruits

both these H3K9MTases explains the functional importance of this mechanism and that IFI16 has evolved to specifically interact/recruit these MTases to establish heterochromatin. Although KD of SUV39H1 and GLP resulted in the induction of KSHV lytic transcripts in BCBL-1 cells, KD of GLP only caused a marginal twofold increase in KSHV genome DNA replication, while KD of SUV39H1 alone had no effect. This may signify that IFI16-mediated recruitment of H3K9me3 is one among the multiple barriers that herpesviruses have to overcome before the successful induction of lytic replication. Other important barriers including H3K27me3 exists and a concerted action of all these viral restriction factors help to establish and maintain latency.

The above suggestion is also supported by the observation that H3K4me3 modification on ORF73 promoter after IFI16 KD is significant (*Figure 1*.F, p-value **). We have previously reported ORF73 mRNA expression under similar conditions after IFI16 KD (*Roy et al., 2016*), where we observed that KD of IFI16 lead to a marginal increase (2-fold) in ORF73 expression in contrast to lytic genes which increased by several folds. This observation can be explained by the fact that the ORF73 promoter is under the concerted regulation of numerous transcription and epigenetic factors including Oct1, GATA-1 Ap1, and RBP-Jκ (*Lan et al., 2005*). In addition, the ORF73 promoter consists of two mRNA start sites – the LANA constitutive (LTc) and the RTA-inducible (LTi) mRNA start sites (*Veeranna et al., 2012*). This makes the ORF73 promoter particularly complex and it is possible that additional factors are involved in keeping the expression of LANA in check after IFI16 KD despite the fact that H3K4me3 is significantly enriched on the ORF73 promoter.

Chromatin marked by H3K9me3 are recognized by the chromo domain protein, HP1α which facilitate heterochromatin formation (*Bannister et al., 2001*; *Lachner et al., 2001*). HP1 proteins serves as a scaffold to recruit other chromatin modifying proteins, including more H3K9 MTases and histone deacetylases (*Bannister et al., 2001*; *Nakayama et al., 2001*; *Lachner et al., 2001*). We therefore reasoned that if IFI16 recruits SUV39H1 and GLP to establish H3k9me3 marks on the KSHV genome, the subsequent recruitment of HP1α should also be dependent on IFI16. Our data showed that depletion of IFI16 resulted in reduced recruitment of HP1α on the KSHV genome (*Figure 9E–G*) and KD of HP1α caused robust induction of KSHV lytic gene transcription. Together, these confirmed that HP1α is important for maintenance of latency and IFI16 is indispensable for its recruitment. Studies of HP1 dynamics have revealed that HP1α is not a stable component of heterochromatin but is highly mobile (*Krouwels et al., 2005*). This dynamicity of HP1α together with the need to epigenetically silence nascent KSHV genomes synthesized as a result of multiplication of latently infected cells helps explain the indispensable nature of IFI16 in HP1α recruitment.

One common factor binding all the identified functions of IFI16 is its DNA binding ability. However, crystal structure of IFI16 HIN domains in complex with B-form dsDNA revealed that it binds dsDNA in a non-sequence specific manner (*Jin et al., 2012*). Therefore, how IFI16 distinguishes between self and non-self DNA and if there is any specific signal that dictates both innate sensing and epigenetic silencing by IFI16 remains unanswered. Further studies are required to determine the possible existence of nuclear macromolecular complex(es) detecting and epigenetically silencing foreign invading DNA. Multiple observations point such a possibility. We have observed that silencing of IFI16 also effects the recruitment of H3K4me3 (*Figure 1C and F*). Therefore, it is possible that a H3K4-specific demethylase is also recruited by IFI16. *Fritsch et al. (2010)* reported that Suv39H1, G9a, GLP, and SETDB1 participate in a multimeric complex. Suv39H1 has been found to interact with the DNA methyltransferase DNMT3B (*Lehnertz et al., 2003*) and HP1α has been reported to recruit DNMT3B. Previous reports have also shown that IFI16 is instrumental in recruiting transcription factors like Sp1 and Oct1 (*Gariano et al., 2012*; *Johnson et al., 2014*). Therefore, the possibility of all these chromatin silencing factors acting in a concerted fashion is plausible.

Our earlier studies have demonstrated that IFI16 is in association with different proteins in the nucleus and mediates the innate immune responses (*Dutta et al., 2015*; *Iqbal et al., 2016*). We identified two IFI16 complexes, namely IFI16-BRCA1 and IFI16-BRCA1-H2B. These complexes recognized the KSHV and HSV-1 genomes soon after their entry into the nucleus, leading into BRCA1-mediated p300 recruitment and acetylation of IFI16 and H2B by p300. IFI16 acetylation resulted in the formation of BRCA1-IFI16-ASC-procaspase-1 inflammasome formation in the nucleus, transport to the cytoplasm, pro-IL-1β cleavage and IL-1β formation. Cytoplasmic transport of acetylated IFI16-H2B-BRCA1 results in the association with cGAS and STING leading into phosphorylation of TBK1 and IRF3, nuclear translocation of p-IRF3 and IFN-β production. Our present study identified a novel epigenetic function of IFI16 and demonstrates that IFI16 is in association with H3K9MTases

SUV39H1 and GLP in the nucleus and IFI16 recruits these MTases to the KSHV genome which sequentially methylates H3K9 to me1/me2 and me3. This serves as a docking site for HP1α which recruits further chromatin compaction factors leading to heterochromatin formation and lytic gene silencing (*Figure 9J*). Thus, our studies uncovered an important paradigm and demonstrated that IFI16-mediated innate immune sensing of foreign viral DNA not only leads to the induction of innate interferon and inflammasome pathways, but also results in the epigenetic silencing of the foreign DNA.

## Materials and methods

### Cells

KSHV-positive PEL cell lines BCBL-1 and BC-3 and KSHV-negative BJAB cells were obtained from the AIDS Malignancy Consortium (AMC) and cultured in RPMI 1640 GlutaMAX (Gibco Life Technologies, Grand Island, NY) supplemented with 10% (v/v) FBS (fetal bovine serum: Atlanta Biologicals) and penicillin-streptomycin (Gibco Life Technologies). TREX-BCBL-1-RTA cells (*Nakamura et al., 2003*) were cultured in the above medium supplemented with hygromycin B (200 µg/ml). TIME (ATCC CRL-4025), a hTERT immortalized dermal microvascular endothelium cell line, was cultured in Vascular Cell Basal Medium (ATCC PCS-100–030), supplemented with Microvascular Endothelial Cell Growth kit-VEGF (ATCC PCS-110–041) and 12.5 ug/ ml blasticidine. U2OS (wt) and U2OS clone 67 - CRISPR IFI16 KO, reported previously by *Johnson et al. (2014)* were cultured in DMEM (Dulbecco's modified Eagle medium, Gibco Life Technologies), supplemented with 10% (v/v) FBS, penicillin-streptomycin and 2 mM L-glutamine (Gibco Life Technologies). 293 T cells were cultured in DMEM supplemented with 10% (v/v) FBS and penicillin-streptomycin and 2 mM L-glutamine (Gibco Life Technologies). All cells were regularly tested for mycoplasma using the Mycoalert kit (Lonza #LT07-218) and were confirmed to be negative.

### KSHV lytic induction and virus production

KSHV lytic cycle was induced in BCBL-1 cells using the 12-O-tetradecanoyl phorbol-13-acetate (TPA; 20 ng/ml). Virion productions and purifications were carried out as per our methods described previously (*Roy et al., 2016*). To quantify the copy number of the virions, KSHV DNA was extracted and quantified by real-time DNA-PCR using primers specific for the KSHV ORF73 gene as described previously (*Roy et al., 2016*). TREX-BCBL-1-RTA cells were induced with doxycycline (DOX, 1 µg/ml).

For de novo KSHV infection, TIME or U2OS cells were washed twice with phosphate buffer saline (PBS), infected with 100 genome copies/cell in serum-free basal medium for 2 hr, washed with PBS and incubated in complete medium from 24 to 96 hr, as indicated.

For the production of BrdU and EdU labeled KSHV, viral DNA was labeled by adding 5-Bromo-2'-deoxyuridine (BrdU) (Thermo Scientific # B23151) 1:100 v/v or 5-ethynyl-2'-deoxyuridine (EdU) (Thermo Scientific #A10044) 10 µM in DMSO into the culture medium of TPA induced BCBL-1 cells. The viral DNA is metabolically labeled during lytic replication. BrdU/EdU was added twice in the culture medium, once on day 1 of TPA induction and again on day 3. The labeled virus from day 5 culture was purified and genome copy number determined as described earlier.

### Lentiviral mediated knockdown of IFI16, SUV39H1, GLP, G9A and HP1a in BCBL1 cells

We used the following human TRC short hairpin RNA (shRNA) constructs (Dharmacon; Horizon Discovery) to co-transfect HEK293T cells along with the lentivirus packaging vectors using the CalPhos mammalian transfection kit (TaKaRa Clontech, Mountain View, CA) as previously described (*Dull et al., 1998*): IFI16 (clones TRCN0000019080, TRCN0000019082, TRCN0000019083), SUV39H1 (clones TRCN0000150622, TRCN0000157251, TRCN0000157285, TRCN0000158270, TRCN0000158337), GLP (clones TRCN0000036054, TRCN0000036055, TRCN0000036056, TRCN0000036057, TRCN0000036058), G9A (clones TRCN0000115667, TRCN0000115668, TRCN0000115669, TRCN0000115670, TRCN0000115671) and HP1α (clones TRCN0000062238, TRCN0000062239, TRCN0000062240, TRCN0000062241). To avoid off-target effects, pools of three or more shRNA were used as stated above. The pLKO.1 empty vector was used as a control (Dharmacon; Horizon Discovery #RHS4080). Culture media were changed after 16 hr of infection,

supernatants containing packaged lentivirus particles collected after 48 hr and filtered through a 0.45 µm filter. Supernatant of all the clones targeting the same gene were pooled together and used to transduce the cells in the presence of polybrene (5 µg/ ml).

## siRNA-mediated knockdown of IFI16, SUV39H1, GLP and G9A in TIME cells

siRNA transfection of TIME cells was performed using the Neon Transfection System (Invitrogen) according to the manufacturer's instructions. Briefly, sub-confluent cells were harvested, washed once with PBS and re-suspended in resuspension buffer R (provided in the kit) at a density of $1 \times 10^7$ cells/ml. Ten microliters of cell suspension was mixed with 100 pmol of respective si-RNA and then microporated at room temperature using a single pulse of 1350 V for 30 ms. Microporated cells suspended in complete medium were kept at 37°C in an atmosphere of 5% CO2. Cells were analyzed for knockdown efficiency by western blotting and/or qRT-PCR. For all the genes, a mixture of 4 siRNA provided as a single reagent (siGenome SMARTpool) was used. Human siGENOME SMART-pool against IFI16, SUV39H1, GLP and G9 were purchased from Dharmacon: Horizon Discovery (catalog # M-020004-01-0010, M-009604-02-0010, M-007065-00-0010 and M-006937-01-0010, respectively). As a negative control, siGENOME Non-Targeting siRNA Pool #2 (Dharmacon: Horizon Discovery # D-001206-14-20) was used.

## Overexpression plasmids and transfection

The IFI16-overexpressing plasmid IFI16-FL (pcDNA3-FLAG) was a gift from Cheryl Arrowsmith (Addgene plasmid 35064) (*Liao et al., 2011*). TrueORFGold clone encoding a C-terminal His/DDK tagged IFI16 was custom generated by Origene. The LANA-1 overexpressing plasmid pCI-neo full-length LANA-1 was described previously (*Paudel et al., 2012*). Cells were transfected using the TransIT-X2 Transfection Reagent (Mirus #MIR 6000) according to the manufacturer's instructions.

## Nuclear protein extraction and co-immunoprecipitation (co-IP)

Nuclear fractions were extracted using a nuclear extraction kit (Active motif #40010) following the manufacturer's instruction. Protein concentrations were estimated using the Pierce BCA protein assay kit (Thermo scientific #23225). All nuclear lysates were treated with benzonase (Sigma # E1014-25KU) for 1 hr before co-IP. Equal amounts of clarified nuclear lysates were resuspended in IP Lysis Buffer (Thermo scientific #87788) supplemented with phosphatase inhibitor cocktail (Thermo scientific #78420) and protease inhibitor cocktail (Thermo scientific #78430) and pre-cleared for 1 hr with 15 µl 50/50 slurry of Protein A and G sepharose beads (GE Healthcare Bio-Science # 17-0469-01 and 17-0618-01, respectively) at 4°C. Following this, the pre-cleared lysates were incubated with respective antibodies and 25 µl 50/50 slurry of Protein A/G sepharose beads over-night at 4°C. The captured immune complexes were washed four times with IP wash buffer (10 mM Tris, pH 7.4, 1 mM EDTA, 1 mM EGTA, 150 mM NaCl, 1% Triton X-100, 0.2 mM sodium orthovanadate and protease inhibitor cocktail), boiled with SDS-PAGE sample buffer, resolved on 4–20% SDS-PAGE, and subjected to western blotting.

The ProFoundTM Pull-Down PolyHis Protein-Protein Interaction Kit (Thermo scientific #21277) was used to pulldown His/DDK tagged IFI16 following the manufacturer's instructions.

## MTT toxicity assays

The MTT cell proliferation assay kit (ATCC# 30–1010K) was used for assessing cellular toxicity following manufacturer's instruction.

## Western blot

Nuclear lysates were prepared as mentioned above. Whole cell lysates were prepared using Pierce IP Lysis Buffer supplemented with a protease inhibitor cocktail for 30 min on ice and then sonicated three times at an amplitude setting of 40 with pulses of 15s on and 10s off on a Qsonica Q700 sonicator. The lysates were clarified by centrifugation at 13,000 X g for 12 min at 4°C. Protein concentrations were estimated and equal concentration of proteins resolved on 4–20% SDS PAGE gels. Resolved gels were blotted onto nitrocellulose membranes at 300 mA for 1.5 hr at 4°C, probed with respective primary antibodies (*Table 1*) overnight at 4°C and then probed with respective HRP-

**Table 1.** List of antibodies used and their dilutions for different applications.

| Antibody | Raised in | Company/catalogue# | Dilution used |
|---|---|---|---|
| IFI16 (1G7) | mouse monoclonal | Santa Cruz Biotechnology (#SC-8023) | WB: 1:700 IP: 1:30 ChIP: 1:30 IFA: 1:100 PLA: 1:50 |
| LANA | Rabbit polyclonal | Raised in-house (Uk183) | WB: 1:1000 IP: 1:50 |
| H3 (D2B12) | Rabbit monoclonal | Cell Signaling Technology (#4620S) | WB: 1:5000 ChIP: 1:50 |
| H3K9me1 | Rabbit polyclonal | Abcam (#ab8896) | WB: 1:1000 |
| H3K9me2 [mAbcam 1220] | Mouse monoclonal | Abcam (#ab1220) | WB: 1:1000 |
| H3K9me3 | Rabbit polyclonal | Active Motif (#39161) | WB: 1:2000 IP: 1:50 ChIP: 1:50 PLA: 1:100 |
| H3K27me3 | Rabbit polyclonal | Active Motif (#39155) | WB: 1:2000 IP: 1:50 ChIP: 1:50 PLA: 1:100 |
| H3K79me3 | Rabbit polyclonal | Abcam (#ab2621) | ChIP: 1:30 |
| H3K4me3 | Rabbit polyclonal | Active Motif (#39159) | WB: 1:2000 IP: 1:50 ChIP: 1:50 PLA: 1:100 |
| H3K36me3 | Rabbit polyclonal | Abcam (#ab9050) | ChIP: 1:30 |
| RNA Pol II CTD [8WG16] | Mouse monoclonal | Abcam (#ab817) | ChIP: 1:50 |
| SUV39H1 (MG44) | Mouse monoclonal | Active Motif (# 39785) | WB: 1:500 IP: 1:30 ChIP: 1:30 IFA: 1:100 PLA: 1:50 |
| SUV39H2 | Goat polyclonal | Novus Biologicals (#NB-100–1140) | WB: 1:500 IP: 1:30 ChIP: 1:30 IFA: 1:100 |
| GLP (B0422) | Mouse monoclonal | Novus Biologicals (# PPB0422-00) | WB: 1:1000 IP: 1:50 ChIP: 1:50 IFA: 1:100 PLA: 1:100 |
| G9A (C-9) | Mouse monoclonal | Santa Cruz Biotechnology (#sc-515726) | WB: 1:500 IP: 1:30 ChIP: 1:30 IFA: 1:70 PLA: 1:50 |
| SETDB1 | Rabbit polyclonal | Novus Biologicals (#NBP2-20322) | WB: 1:500 IP: 1:30 ChIP: 1:30 IFA: 1:70 PLA: 1:50 |
| SETDB2 | Goat polyclonal | Novus Biologicals (#NB100-1137) | WB: 1:500 IP: 1:30 ChIP: 1:30 IFA: 1:70 |
| RIZ1 (N-terminal) | Rabbit polyclonal | Abcam (#ab198792) | WB: 1:500 IP: 1:30 |

*Table 1 continued on next page*

*Table 1 continued*

| Antibody | Raised in | Company/catalogue# | Dilution used |
|---|---|---|---|
| HP1α (GA-62) | Rabbit polyclonal | Cell Signaling Technology (#2616S) | WB: 1:1000<br>IP: 1:50<br>ChIP: 1:50<br>IFA: 1:100<br>PLA: 1:100 |
| EZH2 (EPR20108) | Rabbit monoclonal | Abcam (#191250) | WB: 1:5000<br>IP: 1:100 |
| ASC (TMS1) | Mouse monoclonal | MBL International Corporation (#D086-3) | WB: 1:1000 |
| His-tag (HIS.H8) | Mouse monoclonal | Thermo Fisher Scientific (#MA1-21315)) | WB: 1:5000 |
| β-Tubulin (D66) | Mouse Monoclonal | Sigma-Aldrich (#T0198) | WB: 1:5000 |
| β Actin (AC15) | Mouse monoclonal | Sigma-Aldrich (#A5441) | WB: 1:5000 |
| GAPDH | Rabbit polyclonal | Proteintech (#10494–1-AP) | WB: 1:5000 |
| GFP (GF28R) | Mouse monoclonal | Thermo Fisher Scientific (#MA5-15256) | WB: 1:4000 |
| Lamin B | Rabbit polyclonal | Abcam (#ab16048) | WB: 1:1000 |
| Normal IgG Rabbit | Rabbit | Cell Signaling Technology (#2729) | WB: 1:1000<br>IP: 1:50<br>ChIP: 1:50<br>IFA: 1:100 |
| Mouse IgG2a (MOPC-173) | Mouse | Abcam (#ab18413) | WB: 1:1000<br>IP: 1:50<br>ChIP: 1:50<br>IFA: 1:100 |
| BrdU | Rabbit polyclonal | Rockland antibodies and assays (#600–401 C29) | PLA: 1:100 |
| BrdU (MoBU-1) | Mouse monoclonal | Thermo Fisher Scientific (#B35128) | PLA: 1:100 |
| Anti-mouse HRP | Sheep polyclonal | GE Healthcare (#NA931V) | WB: 1:5000 |
| Anti-rabbit HRP | Donkey polyclonal | GE Healthcare (#NA934V) | WB: 1:5000 |
| Anti-goat HRP | Mouse | Santa Cruz Biotechnology (#sc-2354) | WB: 1:5000 |
| Anti-rabbit IgG Conformation Specific HRP (L27A9) | Mouse monoclonal | Cell Signaling Technology (#5127S) | WB: 1:500 |
| Anti-mouse Light Chain specific HRP | Goat polyclonal | Jackson Immuno Research (#115–035174) | WB: 1:1000 |
| Anti-rabbit Light Chain specific HRP | Mouse monoclonal | Jackson Immuno Research (#211-032-171) | WB: 1:1000 |
| Anti-mouse IgG (H+L)-Alexa fluor 488 | Goat polyclonal | Thermo Fisher Scientific (#A11029) | IFA: 1:500 |
| Anti-rabbit IgG (H+L)-Alexa fluor 488 | Donkey polyclonal | Thermo Fisher Scientific (#A21206) | IFA: 1:500 |
| Anti-goat IgG (H+L)-Alexa fluor 488 | Donkey polyclonal | Thermo Fisher Scientific (#A11055) | IFA: 1:500 |

DOI: https://doi.org/10.7554/eLife.49500.020

conjugated secondary antibodies for detection. Wherever mentioned, light-chain-specific secondary antibodies were used to avoid heavy chain bands in WB of co-IP experiments (*Table 1*). The immunoreactive bands were developed using Super Signal West Pico chemiluminescent substrate (Thermo scientific #34078) or Super Signal West Femto chemiluminescent substrate (Thermo scientific #34095) depending on the signal strength. Blots were developed on a Bio-Rad ChemiDoc XRS+ System and analyzed using the Bio-Rad Image Lab software.

## H3K9me3 activity assay

TIME cells were infected with KSHV (100 DNA copies/cell) for 6 or 24 hr followed by isolation of nuclear fraction as described above. The isolated nuclear fraction was treated with benzonase and immunoprecipitated with anti-IFI16 or control IgG antibodies using the Catch and Release v2.0 Reversible Immunoprecipitation System (Millipore Sigma #17–500). Elution was performed under non-denaturing conditions to keep the associated H3K9 methyltransferase active. Subsequently, the EpiQuik Histone Methyltransferase Activity/Inhibition Assay Kit (H3K9) (Epigentek #P-3003–2) was used to measure the H3K9 methyltransferase activity in the IP eluate following manufacturer's instructions.

## Immunofluorescence assay (IFA)

TIME cells grown on eight-well chamber glass slides were infected with EdU-KSHV (100 DNA copies/cell) for 24 hr, fixed using 4% paraformaldehyde for 15 min, and permeabilized with 0.2% Triton X-100 in PBS for 20 min. Slides were washed, blocked with Image-iT FX signal enhancer (Invitrogen #I36933) for 30 min at 37°C, incubated with primary antibodies for proteins of interest (*Table 1*) for 1 hr at 37°C, washed three times and incubated with corresponding fluorescent dye-conjugated secondary antibodies (*Table 1*). To fluorescent stained EdU labeled viral genome, a CLICK chemistry-based reaction was performed using Click-iTTM EdU Alexa FluorTM 594 imaging kit (Invitrogen #C10339) following the manufacturer's instructions. Slides were mounted using mounting medium containing DAPI and observed either by a Nikon Eclipse 80i microscope or a Keyence BZ-X fluorescence microscope. Images were acquired at 40X magnification and analyzed using image analysis software provided by the respective manufacturers.

## Proximity Ligation Assay (PLA)

TIME, U2OS Wt and U2OS 67 cells grown on eight well chamber glass slides were infected with either EdU or BrdU KSHV (100 DNA copies/cell), and fixed and permeabilized using the same methods as described for IFA. PLA was performed according to the manufacturer's instructions using the following kits and reagents: Duolink In Situ PLA Probe Anti-Rabbit PLUS (Sigma-Aldrich # DUO92002), Duolink In Situ PLA Probe Anti-Mouse MINUS (Sigma-Aldrich # DUO92004), Duolink In Situ Detection Reagents Red (Sigma-Aldrich # DUO92008). Primary antibodies used are listed in *Table 1*. In experiments with anti-BrdU antibodies for PLA, cells were denatured with 4N HCL for 10 min at room temperature after permeabilization with 0.2% Triton X-100. For negative control, isotype matched IgG was used in place of primary antibodies. In experiments where EdU-labeled genome staining was performed along with PLA, slides were subjected to CLICK reaction using Click-iTTM EdU Alexa FluorTM 594 imaging kit as described above immediately after PLA reactions. Experiments where PLA and IFA were performed simultaneously on the same sample, PLA protocol was performed till the ligation step and then the cells were processed for IFA as described above. After binding of the fluorescent dye-conjugated secondary antibodies for IFA, PLA was resumed, and polymerization was performed. Slides were then washed and mounted using a minimal volume of Duolink In Situ Mounting Medium with DAPI (Sigma-Aldrich # DUO82040). PLA signals were detected as distinct fluorescent dots or puncta using either a Nikon Eclipse 80i fluorescence microscope or a Keyence BZ-X fluorescence microscope. Images were acquired at 40X magnification and analyzed using image analysis software provided by the respective manufacturers.

## Edu labeled KSHV chromatin pull down assay

EdU-labeled genome (chromatin) pull down has been described previously (*Dutta et al., 2015*). Briefly, 107 TIME cells were infected with unlabeled or EdU-labeled KSHV (100 DNA copies/cell) for 2 hr, washed and cross-linked using 1% formaldehyde for 10 min at RT. For IFI16 KD condition, cells

were microporated with either siC or siIFI16 72 hr before infection. Unreacted formaldehyde was quenched using 0.125 M glycine for 10 min at RT. Cells were then harvested by trypsinization and permeabilized with 0.2% (v/v) Triton X-100 in PBS for 10 mins on ice and washed with PBS. Following this, Click chemistry was used to covalently couple biotin azide to the EdU genome. For this, the following reagents were added sequentially: 10 mM (+)-Sodium-L-ascorbate, 0.1 mM biotin-TEG azide and 2 mM copper (II) sulfate. Reactions were incubated in the dark for 30 mins at RT following which, reaction was quenched with 10 volumes of 1% (w/v) BSA and 0.5% (v/v) Tween 20 in PBS for 10 mins. Cells were washed three times with PBS and nuclei isolated by incubating in 500 µl CL lysis buffer (50 mM HEPES, pH 7.8, 150 mM NaCl, 0.5% (v/v) NP-40, 0.25% (v/v) Triton X-100, 10% (v/v) glycerol) with protease inhibitors for 10 min at 4°C. The isolated nuclei were then pelleted, washed with 500 µl wash buffer (10 mM Tris-HCL pH 8.0, 200 mM NaCl, 0.5 mM DTT) for 10 min at 4°C, and lysed by resuspension in 500 µl RIPA buffer (10 mM Tris-HCl, pH 8.0, 140 mM NaCl, 1% (v/v) Triton X-100, 0.1% (v/v) Na-Deoxycholate, 0.1% (w/v) SDS) with protease inhibitor cocktail. These were processed for shearing of the chromatin via sonication on ice at an amplitude setting of 40 with pulses of 15 s on and 10 s off for a total of 10 mins on a Qsonica Q700 sonicator. The extract was then clarified by centrifugation at 15,000 x g for 10 min at 4°C and protein content quantitated by the BCA method. 1 mg of the protein extract was pulled-down for 3 hr at 4°C with 50 µl of Dynabeads MyOne Streptavidin T1 (Thermo scientific #65601) which were previously washed with wash buffer, equilibrated with RIPA buffer and blocked overnight at 4°C with 0.5 mg/ml BSA and 0.4 mg/ml pre-sheared salmon sperm DNA to minimize non-specific binding. Beads with bound complexes were then washed three times with wash buffer and subjected to reverse protein-DNA cross-linking and elution of proteins by incubation with 2X Laemmli sample buffer for 10 min at 95°C before SDS-PAGE and WB.

## Measurement of KSHV gene expression by real-time RT-PCR

Total RNA was isolated using the RNeasy minikit (Qiagen #74106) following manufacturer's instructions. On-column DNase digestion was performed by using an RNase-free DNase set (Qiagen #79254). Concentration of the extracted RNA was estimated using a NanoDrop spectrophotometer (Thermo Scientific), and 1 µg RNA was reverse transcribed by using the High-Capacity cDNA reverse transcription kit (Applied Biosystems #4368814) with random primers, according to the manufacturer's instructions. For real-time quantitative reverse transcription-PCR (qRT-PCR) in PEL cells, the synthesized cDNA was used as a template with Power SYBR Green PCR Master Mix (Applied Biosystems #4367659) on an ABI Prism 7500 detection system (Applied Biosystems). All RNA levels were normalized to β-actin mRNA levels and calculated as the delta-delta threshold cycle (ΔΔCT). All primers used have been described previously (*Roy et al., 2016*). In KSHV infected TIME cells, ORF 73 and ORF 50 mRNA were quantified by real time RT-PCR using gene-specific TaqMan primers, probes and standards that has been described previously (*Krishnan et al., 2004*). The TaqManTM RNA-to-CTTM 1-Step Kit (Applied Biosystems #4392938) was used following manufacturer's instructions. Viral mRNA expressions were normalized to cellular internal control, RNaseP using manufacturer's instruction (Applied Biosystems #4403328).

## Chromatin immunoprecipitation

Chromatin shearing for ChIP was performed by using the truChIP Chromatin Shearing kit (Covaris #520154) following manufacturer's instructions on a Covaris ME220 focused ultrasonicator. After chromatin shearing, Triton X-100 and NaCl in the sheared lysate were adjusted to final concentrations of 1% and 150 mM, respectively. Shearing efficiencies were evaluated by using a 2100 Bioanalyzer instrument and the Agilent High Sensitivity DNA Kit (Agilent Technologies #5067–4626) following manufacturer's instructions. The fragment size was ensured to be between 200 bps and 500 bps. For immunoprecipitation, 10 µg sheared chromatin was immunoprecipitated with 2 µg desired antibody or ChIP-grade control IgG (*Table 1*) overnight at 4°C. The chromatin-antibody complex was pulled down with ChIP-grade protein G magnetic beads (Active Motif #104502) for 2 hr at 4°C. The immunoprecipitated complex was then washed three times with low-salt and once with high-salt wash buffers (Cell Signaling Technology #14231S). To elute the chromatin, the beads were incubated in ChIP elution buffer (Cell Signaling Technology #14231S) at 65°C for 30 min on a ThermoMixer (1,200 rpm). Following this step, the eluted chromatin was incubated with NaCl and

proteinase K for 2 hr at 65°C to remove all proteins and reverse the cross-linking. DNA was purified by using the ChIP DNA Clean and ConcentratorTM kit (Zymo Research #D5205). The ChIP-enriched DNAs were quantitated by real-time quantitative PCR (qPCR) using Power SYBR green PCR master mix (Applied Biosystems #4367659) and primers described previously (*Roy et al., 2016*). ChIP enrichment was calculated as relative to input chromatin (% input) and expressed as fold enrichment over control (shC or siC) ChIP.

## Statistical analysis

Data are expressed as means ± standard deviations (SD) of results from at least three independent experiments (n ≥ 3), and statistical significance was calculated by using the two-tailed Student t test. *=p < 0.05; **=p < 0.01 and ***=p < 0.001.

# Acknowledgements

This study was supported in part by Public Health Service grant CA 180758 and USF start-up fund to Bala Chandran.

# Additional information

## Funding

| Funder | Grant reference number | Author |
| --- | --- | --- |
| Public Health Institute | CA 180758 | Bala Chandran |
| University of South Florida | Start-up fund | Bala Chandran |

The funders had no role in study design, data collection and interpretation, or the decision to submit the work for publication.

## Author contributions

Arunava Roy, Conceptualization, Data curation, Formal analysis, Validation, Investigation, Visualization, Methodology, Writing—original draft; Anandita Ghosh, Data curation, Visualization, Methodology; Binod Kumar, Data curation, Methodology; Bala Chandran, Conceptualization, Resources, Formal analysis, Supervision, Funding acquisition, Validation, Investigation, Visualization, Methodology, Writing—original draft, Project administration, Writing—review and editing

## Author ORCIDs

Arunava Roy https://orcid.org/0000-0002-8486-0539
Bala Chandran https://orcid.org/0000-0002-5319-8714

## Decision letter and Author response

Decision letter https://doi.org/10.7554/eLife.49500.023
Author response https://doi.org/10.7554/eLife.49500.024

# Additional files

## Supplementary files

• Transparent reporting form DOI: https://doi.org/10.7554/eLife.49500.021

## Data availability

All data generated or analysed during this study are included in the manuscript and supporting files.

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
