## [Decision Letter]

Thank you for submitting your article "IFI16, an innate DNA sensor, mediates epigenetic silencing by its association with H3K9methyltransferase SUV39H1 and GLP" for consideration by *eLife*. Your article has been reviewed by three peer reviewers, and the evaluation has been overseen by a Reviewing Editor and Anna Akhmanova as the Senior Editor. The following individual involved in review of your submission has agreed to reveal their identity: Ravi Mahalingam (Reviewer #2).

The reviewers found the manuscript to be important, thorough, and well-written. The reviewers have discussed the reviews with one another and the Reviewing Editor has drafted this decision to help you prepare a revised submission.

Summary:

This study aims to define the mechanism by which the intrinsic DNA sensor, IFI16, suppresses transcription of the lytic genes encoded by KSHV. Previous work by this group and others has defined IFI16 as a restriction factor for herpesvirus lytic replication including as an inducer of the inflammasome and also as a repressor of lytic gene expression. Other groups have defined the epigenetic landscape of KSHV infected cells in latency and upon reactivation implicating H3K9me3 at two large regions of late lytic genes in the genome. The question being addressed by this study is precisely which histone marks are regulated by IFI16 through which methyltransferases and how this impacts repression of lytic gene expression. The authors take a comprehensive approach to this problem with many specific histone marks, methyltransferases, and cellular models under study.

Title: The title does not have the word "virus" in it anywhere, and therefore this looks like a chromatin paper. Would you consider changing it to: "IFI16 mediates epigenetic silencing of herpesvirus genomes by its association with H3K9 methyltransferases SUV39H1 and GLP"?

Essential revisions:

1) What are the effects of IFI16, SUV39H1/GLP, or HP1a loss on KSHV viral DNA replication and particle formation? Is the IFI16/H3K9me3 barrier an initial barrier to RTA expression, but another H3K27me3 barrier exists (e.g. Toth data) for full lytic reactivation including DNA replication, late gene expression, and particle production?

2) What is the impact of IFI16, SUV39H1/GLP, and HP1a loss on RTA-induced lytic reactivation? Related to the question above, does RTA over-expression mitigate IFI16 mediated suppression?

3) Subsection “Knockdown of IFI16 causes prominent changes in the deposition of H3 lysine methylation marks and RNA Pol II on KSHV promoters”, second paragraph. Why do the authors artificially define the significance as above 2-fold or below 0.5-fold, and not depend on the statistical analysis in their experiments? It seems that H3K4me3 modification on ORF73 promoter after IFI16 knock-down is quite significant (Figure 1C). In addition, H3K4me3 is recognized as a histone marker to identify active gene promoter and promote gene expression. Did they determine the level of ORF73 expression side by side in a similar experiment? The p-values should support the findings and the 2-fold lines should be removed.

4) The statistics in ChIP experiments in Figure 1C are confusing. For example, the large change in pORF73 H3K4me3 upon IFI16 loss with minimal error has "ns" for non-significant above it. Likewise, the H3K27me3 and H3K36me3 bars with high variation have "*" for p<0.05. And the pvIRF2 H3K27me3 with no change has "**" for p<0.01. While it is conceivable that these statistical tests do not correlate with the error and effect sizes, it seems that there may be some miscalculations or misplacements of these error notations.

5) There are no quantitative assessments of the pulldown efficiencies of the IFI16 interactions to assess their significance.

6) There is no quantitation of the PLA assays. This is a significant issue because conclusions can only be drawn by assessing the PLA in one or two cells. These data would be strengthened with the inclusion of data from at least 50 cells and appropriate statistical analysis.

7) There is no quantitation of the H3-MTase/EdU co-localization data in Figure 2F. Since the EdU signal is so extensive, it is hard to evaluate the relative co-localization of these enzymes with genomes. It might be useful to do this experiment with a lower MOI of EdU labeled virus and also to perform quantitation on the co-localization of these signals to better understand the specificity of these interactions.

---

## [Author Response]

Title: The title does not have the word "virus" in it anywhere, and therefore this looks like a chromatin paper. Would you consider changing it to: "IFI16 mediates epigenetic silencing of herpesvirus genomes by its association with H3K9 methyltransferases SUV39H1 and GLP"?

Yes and we thank for this excellent suggestion.

However, we request the editors to allow us to retain the “a nuclear innate immune DNA sensor” in the title as it conveys the initial function of IFI16 that was identified, namely, its function as a nuclear innate sensor of episomal nuclear DNA.

The new title we would like to suggest is as follows:

“IFI16, a nuclear innate immune DNA sensor, mediates epigenetic silencing of herpesvirus genomes by its association with H3K9 methyltransferases SUV39H1 and GLP.”

Essential revisions:1) What are the effects of IFI16, SUV39H1/GLP, or HP1a loss on KSHV viral DNA replication and particle formation? Is the IFI16/H3K9me3 barrier an initial barrier to RTA expression, but another H3K27me3 barrier exists (e.g. Toth data) for full lytic reactivation including DNA replication, late gene expression, and particle production?

We thank the reviewers for these helpful suggestions.

We have previously reported the effect of IFI16 KD on KSHV DNA replication and particle formation in BCBL-1 cells (Roy et al., 2016 – Figure 2). There, we had observed that IFI16 KD resulted in 5-fold increase in the intracellular viral genome copy numbers, and the DNase resistant packaged virion genome copy numbers increased by 2.6-fold.

We have now included this information in the second paragraph of the Introduction section.

For this revised manuscript, we have now performed new experiments to address the reviewer’s question about the effects of SUV39H1, GLP, G9A (new Fig. 8H) and HP1α (new Fig. 9I) loss on KSHV viral DNA replication. "

These new observations are discussed in the first paragraph of the subsection “H3K9 MTase SUV39H1 and GLP are essential for KSHV gene regulation but not G9A” and in the second paragraph of the subsection “Recruitment of Heterochromatin Protein 1α (HP1α) on the KSHV genome is 513 dependent on IFI16 mediated H3K9-trimethylation”, respectively.

In this new experiment, we have KD SUV39H1, GLP, G9A and HP1α in BCBL-1 cells and quantitated the KSHV genomic DNA copy number representing KSHV replication after 2 and 4 days of KD. We observed that KD of GLP resulted in a 2-fold increase in the intracellular viral genome copy numbers after 4 days of shRNA treatment, while KD of SUV39H1 and G9A MTases were not sufficient to induce lytic cycle replication. KD of HP1α also, failed to induce lytic reactivation.

In light of this new data, we agree with the reviewers that IFI16 mediated recruitment of H3K9me3 is one among the multiple barriers that herpesviruses have to overcome before the successful induction of lytic replication. Other important barriers including H3K27me3 exists and a concerted action of all these viral restriction factors help to establish and maintain latency.

We have included this information in the fourth paragraph of the Discussion section.

2) What is the impact of IFI16, SUV39H1/GLP, and HP1a loss on RTA-induced lytic reactivation? Related to the question above, does RTA over-expression mitigate IFI16 mediated suppression?

We have addressed this question by performing new experiments in the TREX-BCBL-1 RTA cells.

TRExBCBL1-Rta cells carry latent KSHV genomes and an epitope-tagged KSHV lytic cycle switch replication and transcription activator (RTA) protein cassette under the control of a tetracycline-inducible promoter. Treatment with doxycycline induces the overexpression of RTA, which in turn induces the KSHV lytic reactivation.

We induced the TREXBCBL-1 cells with doxycycline and studied the abundance of IFI16, SUV39H1, GLP, and HP1α on the KSHV genome by ChIP at 1, 2, 3 and 4 days post-induction (new Figure 8A and 9H). These new results demonstrate that over-expression of RTA does mitigate the IFI16 mediated recruitment of SUV39H1, GLP, and HP1α.

The abundance of SUV39H1 on the KSHV genome decreased to about 25% of that of uninduced cells on day 2 post-induction and remained at that level until day 4. GLP, on the other hand showed a more drastic decrease and was reduced to less than 10% of the uninduced level by day 1 post-induction. Abundance of IFI16 decreased more gradually and reduced to less than 20% by day 4 which could be also due to the specific IFI16 degradation during KSHV lytic reactivation as demonstrated by us before (Roy et al., 2016 – Figure 7).

In addition, HP1α abundance dropped by about 60% during reactivation.

These new observations are discussed in the last paragraph of the subsection “PLA in IFI16 KO cells confirms that IFI16 recruits SUV39H1 and GLP onto the KSHV genome resulting in the deposition of H3K9me3 but not H3K27me3” and in the second paragraph of the subsection “Recruitment of Heterochromatin Protein 1α (HP1α) on the KSHV genome is dependent on IFI16 mediated H3K9-trimethylation”.

3) Subsection “Knockdown of IFI16 causes prominent changes in the deposition of H3 lysine methylation marks and RNA Pol II on KSHV promoters”, second paragraph. Why do the authors artificially define the significance as above 2-fold or below 0.5-fold, and not depend on the statistical analysis in their experiments? It seems that H3K4me3 modification on ORF73 promoter after IFI16 knock-down is quite significant (Figure 1C). In addition, H3K4me3 is recognized as a histone marker to identify active gene promoter and promote gene expression. Did they determine the level of ORF73 expression side by side in a similar experiment? The p-values should support the findings and the 2-fold lines should be removed.

We thank the reviewers for these constructive suggestions. As suggested, we have removed the 2-fold lines and now support our data with appropriate p-values.

It has been rightly pointed out by the reviewers that H3K4me3 modification on ORF73 promoter after IFI16 KD is significant (p-value **).

ORF73 mRNA expression under similar conditions after IFI16 KD has been reported by us previously (Roy et al., 2016 – Figures 1 and 2). There, we had observed that KD of IFI16 lead to a marginal increase (2 fold) in ORF73 expression in contrast to lytic genes which increased by several folds. This observation can be explained by the fact that the ORF73 promoter is under the concerted regulation of numerous transcription and epigenetic factors including Oct1, GATA-1 Ap1, and RBP-Jκ (Lan et al., 2005). In addition, the ORF73 promoter consists of two mRNA start sites – the LANA constitutive (LTc) and the RTA-inducible (LTi) mRNA start sites (Veeranna et al., 2012). This makes the ORF73 promoter particularly complex and it is possible that additional factors are involved in keeping the expression of LANA in check after IFI16 KD despite the fact that H3K4me3 is significantly enriched on the ORF73 promoter.

This is included in the sixth paragraph of the Discussion section.

4) The statistics in ChIP experiments in Figure 1C are confusing. For example, the large change in pORF73 H3K4me3 upon IFI16 loss with minimal error has "ns" for non-significant above it. Likewise, the H3K27me3 and H3K36me3 bars with high variation have "*" for p<0.05. And the pvIRF2 H3K27me3 with no change has "**" for p<0.01. While it is conceivable that these statistical tests do not correlate with the error and effect sizes, it seems that there may be some miscalculations or misplacements of these error notations.

We greatly thank the reviewers for pointing out this mistake. We have re-calculated the p-values and rectified the mistakes in placement accordingly. However, for the pvIRF2 H3K27me3 data point, re-calculation still yielded the result as significant.

5) There are no quantitative assessments of the pulldown efficiencies of the IFI16 interactions to assess their significance.

Co-IP is a standard and robust method to detect protein-protein interactions. Co-IP followed by western blotting is not a quantitative method and our objective in this study is to attain qualitative information whether IFI16 interacts with our protein of interests or not. Moreover, antibodies against all these proteins are very diverse with varying binding efficiencies and therefore achieving a quantitative assessment of their pulldown efficiencies will be challenging if not impossible. We have assessed the specificity of the interaction by performing IgG IPs and confirmed the IFI16 mediated recruitment of SUV39H1 and GLP by methods other that co-IP like EdU labeled genome pulldowns (Figure 5A) and His-tagged IFI16 pulldowns (Figure 4C), both of which are antibody independent.

In addition, interactions and the specificities were confirmed by PLA as well.

6) There is no quantitation of the PLA assays. This is a significant issue because conclusions can only be drawn by assessing the PLA in one or two cells. These data would be strengthened with the inclusion of data from at least 50 cells and appropriate statistical analysis.

PLA detects protein-protein interactions at <40 nm compared to the ~200nm of IFA. PLA detects the endogenous protein levels, detects the weak and transient interactions that might not survive the co-IP manipulations, and gives spatial and subcellular resolutions. PLA will not work if two proteins are not in close proximity and PLA dots represent the close proximity (interactions) of two proteins being examined. Hence, it is a positive or negative testing in our system.

In the earlier submission, we have shown the specificity of the PLA reaction between IFI16 and MTases by including an IFI16 + IgG PLA (Figure 5C, bottom panel) which does not show any positive PLA dots.

Similarly, absence of PLA signal between BrdU KSHV+H3K9me3, BrdU KSHV+SUV39H1 and BrdU KSHV+GLP in uninfected TIME cells (UI) were shown to confirm that non-specific PLA interactions were absent in our system (Figure 7—figure supplement 1).

We have now included new figures showing the absence of significant PLA signal between BrdU labelled KSHV genome and IgG (Figure 6—figure supplement 1) and IgG+IgG control PLAs (Figure 5—figure supplement 1) (subsection “IFI16 interacts with H3K9MTase SUV39H1 and GLP and recruits them on to the KSHV genome”, second paragraph and subsection “PLA in IFI16 KO cells confirms that IFI16 recruits SUV39H1 and GLP onto the KSHV genome resulting in the deposition of H3K9me3 but not H3K27me3”, first paragraph).

Additionally, we have now included the source un-cropped images for all PLA data in this manuscript as Figure 5—figure supplements 2 and 3, Figure 6—figure supplements 2 and 3, and Figure 7—figure supplements 2 and 3. This shows multiple cells for all experiments and confirms that these results are significant. We have also shown IFI16 transfected and un-transfected cells in the same field in Figure 7. However, we regret that we do not have images for 50 cells for all experimental conditions. We believe that these supportive data together are convincing evidence that the recruitment of H3K9me3, SUV39H1 and GLP are IFI16 dependent.

7) There is no quantitation of the H3-MTase/EdU co-localization data in Figure 2F. Since the EdU signal is so extensive, it is hard to evaluate the relative co-localization of these enzymes with genomes. It might be useful to do this experiment with a lower MOI of EdU labeled virus and also to perform quantitation on the co-localization of these signals to better understand the specificity of these interactions.

We thank the reviewers for their comment. We have deconvoluted the EdU signal (red) in Figure 2F to distinctly visualize KSHV genomes in the nucleus and now we believe the co-localization of the different MTases with the KSHV genome is more prominent.

This improved figure is now included in place of the old Figure 2F.